# Cell state-dependent allelic effects and contextual Mendelian randomization analysis for human brain phenotypes

Alexander Haglund [1,18], Verena Zuber [2,3,4,18], Maya Abouzeid[1], Yifei Yang[1], Jeong Hun Ko[1], Liv Wiemann[1], Maria Otero-Jimenez[1], Louwai Muhammed[5], Rahel Feleke[1], Alexi Nott [1,4], James D. Mills[6,7,8], Liisi Laaniste [1], Djordje O. Gveric[1], Daniel Clode[1], Ann C. Babtie [9], Susanna Pagni[6,7], Ravishankara Bellampalli[6,7], Alyma Somani[6], Karina McDade [10], Jasper J. Anink[8], Lucia Mesarosova[8], Nurun Fancy [1,4], Nanet Willumsen [1,4], Amy Smith[1,4], Johanna Jackson [1,4], Javier Alegre-Abarrategui [1], Eleonora Aronica [8,11,19], Paul M. Matthews [1,4,19], Maria Thom[6,19], Sanjay M. Sisodiya[6,7,19], Prashant K. Srivastava[12,19], Dheeraj Malhotra[13,14,19], Julien Bryois [13,19], Leonardo Bottolo [15,16,17,19] ✉ & Michael R. Johnson [1,19] ✉

Gene expression quantitative trait loci are widely used to infer relationships between genes and central nervous system (CNS) phenotypes; however, the effect of brain disease on these inferences is unclear. Using 2,348,438 single-nuclei profiles from 391 disease-case and control brains, we report 13,939 genes whose expression correlated with genetic variation, of which 16.7–40.8% (depending on cell type) showed disease-dependent allelic effects. Across 501 colocalizations for 30 CNS traits, 23.6% had a disease dependency, even after adjusting for disease status. To estimate the unconfounded effect of genes on outcomes, we repeated the analysis using nondiseased brains ($n = 183$) and reported an additional 91 colocalizations not present in the larger mixed disease and control dataset, demonstrating enhanced interpretation of disease-associated variants. Principled implementation of single-cell Mendelian randomization in control-only brains identified 140 putatively causal gene–trait associations, of which 11 were replicated in the UK Biobank, prioritizing candidate peripheral biomarkers predictive of CNS outcomes.

Translating genome-wide association study (GWAS) loci to therapies requires knowledge of the causal genes, their directionality of effect and the cell types in which they act. In this study, we aimed to infer these relationships by implementing a principled approach to Mendelian randomization (MR) using single-cell expression quantitative trait loci (eQTL) as genetic anchors[1,2]. MR is a statistical and methodological framework for inferring putatively causal effects rooted in the naturally randomized allocation of genetic variants instrumenting exposures, such as the level of expression of a gene[3].

Previous research mapping eQTLs at the single-cell level has high-lighted dynamic cell state-dependent influences on gene regulation[4], and prior work using bidirectional MR has suggested that most disease-associated gene expression changes occur as a consequence of disease rather than as a cause[5]. For central nervous system (CNS) phenotypes, disease-based brain tissue samples have contributed to several studies reporting putatively causal associations between tran-script levels and CNS outcomes at both bulk-tissue[6] and single-cell[7,8] levels, but the potential impact of using diseased samples for causal

inference analysis has been largely unassessed. While the use of diseased brain samples can increase sample size and statistical power, where the disease itself causes gene expression changes, their use might obscure pathways relevant to disease etiology or prioritize spurious associations from reverse causation[5].

Here we highlight dynamic allelic effects on gene expression arising from the presence of brain disease and show how these influence the results of downstream analyses that seek to inform biological mediators of brain disease. Using a unique set of control brains with no history of CNS disease and normal neuropathology, we show that a principled approach to MR using control brain samples can provide estimates of the direction of an effect of a gene on CNS outcomes unconfounded by disease state and enhance the interpretation of GWAS loci.

## Results

### Datasets and eQTL discovery

We analyzed single-nuclei gene expression data (single-nuclei RNA sequencing (snRNA-seq)) based on postmortem brain tissue samples from 409 genotyped adult donors. Following quality control, data harmonization and cell-type annotation, we retained 2,348,438 single nuclei from 391 individuals (median nuclei per individual = 4,391 and mean = 6,006; Supplementary Fig. 1). The donors consisted of 183 participants with no history of brain disease and normal appearances of the brain on neuropathological examination and 208 participants who had died with a documented neurological diagnosis. Single-cell-type cis-eQTLs were identified using a linear model implemented in Matrix-EQTL[9] as previously described[8]. This analysis was based on residual gene expression after adjusting for clinical covariates (age at death, sex, postmortem interval (PMI), disease status and sample source) and optimized principal components (PCs), all treated as fixed covariates for each cell type. In total, we tested 5.20 million single-nucleotide polymorphisms (SNPs) for cis-gene regulation in eight brain cell types (excitatory neurons, inhibitory neurons, astrocytes, microglia, oligodendrocytes, oligodendrocyte precursor cells (OPCs), endothelial cells and pericytes; Fig. 1a and Supplementary Fig. 2).

Across the 391 mixed disease-case and control participants, we captured 1.82 million single-cell-type cis-eQTLs at false discovery rate (FDR) < 5%, representing one or more regulatory SNP (eSNP) for 13,939 unique genes (30,027 eGenes in aggregate; Fig. 1b and Supplementary Table 1). Of these, 5,454 (39.1%) were identified in only a single cell type (Supplementary Fig. 3). Only eight genes shared their genetic regulation across all eight cell types. SNP–transcript associations were distributed above the expected uniform distribution (Supplementary Fig. 4). The number of cells profiled for a particular cell type linearly correlated with the number of eGenes for that cell type (Pearson correlation $r = 0.92$, $P = 1.14 \times 10^{-3}$; Supplementary Fig. 5). As previously reported[7], we observed an enrichment of cis-eQTLs around the target gene transcription start site (Supplementary Fig. 6). Most cis-eQTLs (72.9–88.7% depending on cell type) replicated (FDR < 5%) in a large (6,523 participants) cortex tissue-level eQTL study[10], of which 90.0–98.3% had the same direction of effect (Fig. 1c). Notably, analysis of cis-eQTLs at a single-cell-type level identified 4,898 more eGenes compared to cis-eQTL discovery using an equivalently sized 'bulk' tissue analysis based on aggregating counts across all cell types (Fig. 1d).

### Assessment of disease status on cis-eQTLs

Because the primary objective of this work was to identify genetically regulated exposures that influence disease risk, we first aimed to understand the effect of brain disease on the genetic regulation of gene expression in the human brain. In particular, for datasets consisting of mixed disease-case and control samples, which has been the standard experimental design to date[7,8], whether the usual approach of adjusting gene expression for disease status adequately accounts for the effects of disease on brain gene expression. Using our combined dataset of 391 disease and control participants and following the methodology

discussed in ref. 11, we refitted each of the discovered eGenes for each cell type against its top regulatory SNP (eSNP) based on the eQTL P value using linear mixed effects (LME) models (Fig. 2a). First, we evaluated the null model ($M_0$), where the expression of each gene was fitted against clinical covariates (age at death, sex, PMI and disease status as fixed effects). To account for diagnosis-specific variation within each sample source, we used a nested random effect for 'disease status' on 'sample source'. These effects are important to consider due to potential bias from the brain bank itself and from the clinical diagnosis, which might vary between pathologists and cannot be captured by adding sample source as a fixed effect. We then tested the SNP–gene model ($M_1$) with a similar model configuration, except with the addition of the genotype to model an eQTL association. If the $M_1$ was a better fit than $M_0$, we considered this a 'pass' and a validation of the initial genome-wide eQTL result. Across all cell types, all but one association (rs10762316–HK1 expression in inhibitory neurons) was statistically more adequate than the null model (without an SNP). Furthermore, eQTL P values from the $M_1$ model correlated well with the discovery P values, suggesting that the inclusion of nested random effects has a negligible impact on the overall associations (Supplementary Fig. 7).

We then tested whether any of the cis-eQTL associations were better modeled with an interaction term between the disease diagnosis ($D$) and the genotype ($G$) ($M_2$). Depending on cell type, we found that an average of 16.7–40.8% of eQTL associations had a significant interaction with disease (that is, a likelihood ratio test (LRT) q value < 0.05 in favor of the interaction model $M_2$; Fig. 2b,c and Supplementary Table 2). Specific examples of disease-dependent allelic associations with gene expression are shown for microglial, astrocyte, oligodendrocyte and excitatory neuron eQTLs in Fig. 2d–g, respectively. In the case of the association between rs117934759 and PTPN12 expression in microglia (Fig. 2d; LRT−$P = 1.72 \times 10^{-16}$, $q = 4.14 \times 10^{-14}$), the eQTL association was strongly influenced by Alzheimer's disease (AD) samples ($AD_{interaction}$, $P = 1.11 \times 10^{-10}$). The association between rs6538127 and NAV3 expression in astrocytes (Fig. 2e; LRT−$P = 1.23 \times 10^{-6}$, $q = 4.93 \times 10^{-5}$) was influenced by both multiple sclerosis ($MS_{interaction}$ $P = 2.02 \times 10^{-5}$) and AD samples ($AD_{interaction}$ $P = 2.94 \times 10^{-3}$), while rs7932358 and ARHGAP20 expression in oligodendrocytes (Fig. 2f; LRT−$P = 4.75 \times 10^{-8}$, $q = 2.10 \times 10^{-6}$) was influenced by all three disease states (Parkinson's disease (PD)$_{interaction}$, $P = 3.57 \times 10^{-4}$; $MS_{interaction}$, $P = 1.39 \times 10^{-3}$; $AD_{interaction}$, $P = 8.96 \times 10^{-6}$). Associations were also observed across disease-case and control categories, as in the example of rs60935857 and ZNF880 expression in excitatory neurons (Fig. 2g; PD$_{interaction}$, $P = 2.24 \times 10^{-6}$; control, $P = 2.60 \times 10^{-15}$). In most cases, the majority of interactions were influenced by a single disease (4,004 of 6,663 total interactions, 60.1%), and a lesser proportion of interactions were influenced by all three diseases (506, 7.6%). Where interactions were influenced by a single disease, PD was the most common. Microglia were the only cell type where the majority of interactions were driven by AD cases (Supplementary Fig. 8), highlighting the impact of Alzheimer's pathology on microglial gene regulation[12].

Overall, these results are consistent with the interpretation that adjusting for disease status and random effects does not adequately account for the influence of brain disease on genetic regulation of gene expression in the human brain. The consequences of this are important because the relationship between allelic effects on gene expression and disease is critical to the interpretation of eQTL pathogenicity[4].

### Genetic colocalization analysis

Colocalization is a statistical method that seeks to identify biological mediators of disease by assessing whether exposures, such as the level of expression of a gene in a particular cell type and a clinical phenotype, share a common causal variant[13,14]. To explore the impact of brain disease on colocalization, we integrated single-cell-type cis-eQTLs derived from the full dataset of mixed disease ($n = 208$) and control

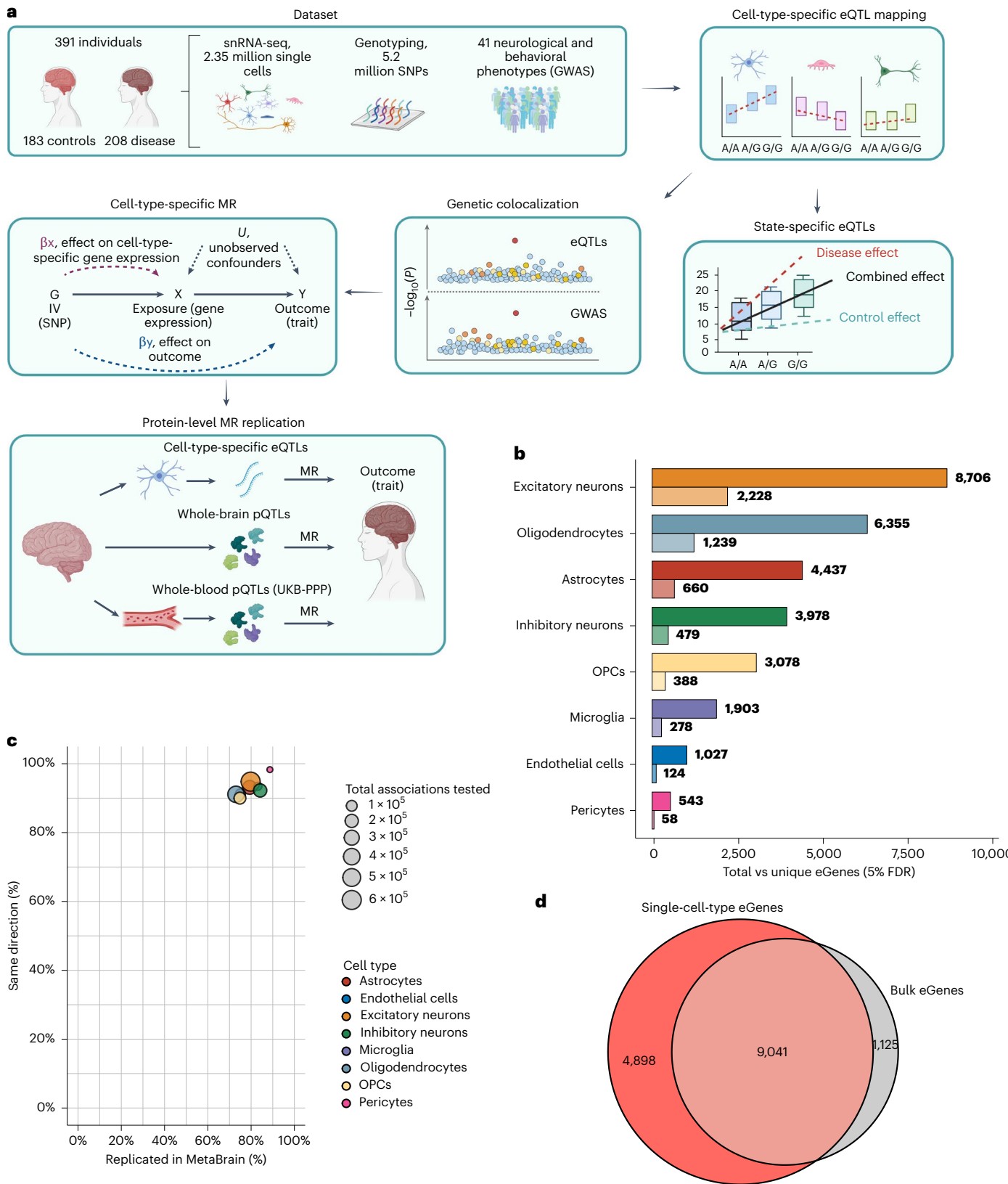

**Fig. 1 | Study design and eQTL analysis. a**, Overview of study workflow, created using BioRender.com. **b**, Total eGenes per cell type (top, darker bar) versus eGenes unique to that cell type (bottom, lighter bar). **c**, Replication of single-cell-type *cis*-eQTLs (number of eQTLs per cell type indicated by bubble size) using *cis*-eQTLs from human cortex bulk RNA-seq dataset (MetaBrain) for both replication (*x* axis) and directionality (*y* axis). **d**, Overlap of single-cell-type eGenes with eGenes from the same single-cell dataset but with counts aggregated across cells to simulate an equivalently sized 'bulk-tissue' gene expression dataset (pseudobulk).

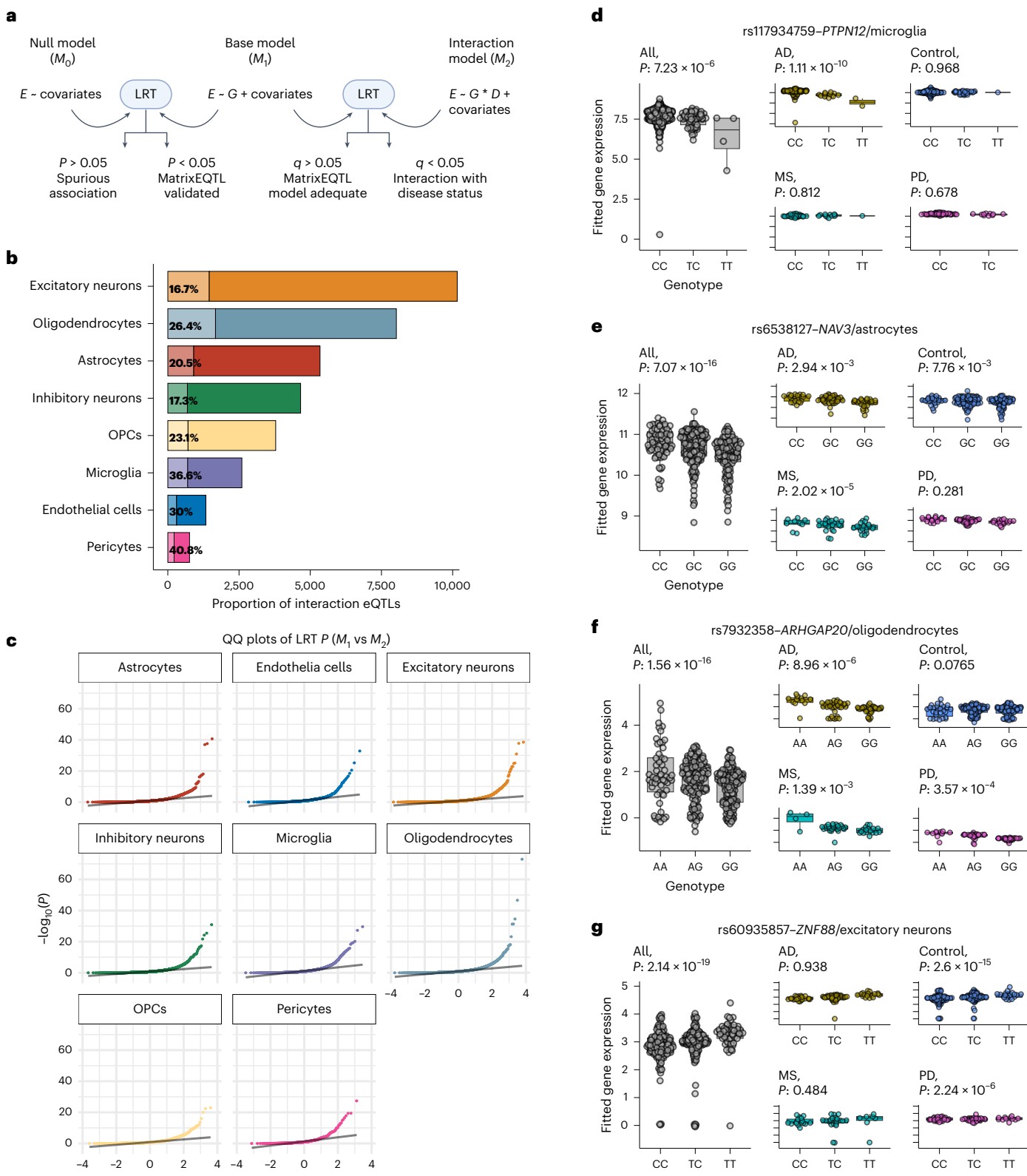

(n = 183) brains (following adjustment for disease status as previously implemented by others[7,8]) with GWAS data from 41 distinct neurological, psychiatric, behavioral and structural brain phenotypes (listed in Supplementary Table 3). We applied COLOC[13] to all chromosomal regions containing a genome-wide significant ($P < 5.0 \times 10^{-8}$) association with a phenotype, based on a 1 Mb window around the lead GWAS SNP.

In total, we identified 501 colocalizations at PP.H4 > 0.8 between the genetic regulation of a gene in a particular cell type (gene/cell-type pair) and genetic risk to one or more of 30 CNS phenotypes (gene/cell-type/trait triplets; Fig. 3a and Supplementary Table 4). The majority of colocalizations (74.4%) mapped to a single cell type. In total, 61 of 249 unique genes (24.5%) colocalized with one (or more) phenotype across multiple cell types, for example, *ICA1L* and CNS white matter

**Fig. 2 | Modeling of disease–interaction *cis*-eQTLs. a**, Overview of the statistical framework. For each single-cell-type *cis*-eQTL tested, we first assessed a null model ($M_0$), testing the association of the gene with clinical and technical covariates. We then tested whether the base model ($M_1$), which includes the use of genotype to model the observed SNP–transcript association, was better suited compared to the null model. We repeated this comparison between the base model and the disease–interaction model ($M_2$), testing whether the use of an interaction term on diagnosis was more appropriate. **b**, Percentage of *cis*-eQTLs with a significant disease interaction (that is, *q* value < 0.05 in favor of the interaction model $M_2$) for each cell type. **c**, QQ plots of observed versus expected LRT *P* values calculated on $M_1$ versus $M_2$, for each cell type, showing significant

deviation from the expected distribution. **d–g**, Examples of single-cell-type *cis*-eQTLs from the full cohort (*n* = 391) where the SNP–gene association has a significant disease interaction for microglia (**d**), astrocytes (**e**), oligodendrocytes (**f**) and excitatory neurons (**g**). The *P* values for 'all' represent the *t* statistic for the $M_1$ models, whereas the *P* values for AD, PD and MS represent the *P* value from the interaction with genotype within the $M_2$ model. The 'control' *P* value represents the effect of genotype on expression after accounting for interaction effects. Elements of the boxes show the center line (median), box limits represent upper and lower quartiles and whiskers represent upper and lower quartils ±1.5× IQR. All data points have been included. QQ, quantile-quantile plots; IQR, interquartile range.

hyperintensities (WMH), which colocalized in astrocytes (PP.H4 = 0.88), excitatory neurons (PP.H4 = 0.89), inhibitory neurons (PP.H4 = 0.85) and OPCs (PP.H4 = 0.89). There was a strong correlation between the number of colocalizations for a particular phenotype and the number of genome-wide significant GWAS loci for that phenotype (Pearson correlation *r* = 0.93, *P* = 2.52 × 10$^{-18}$; Supplementary Fig. 9). Repeated analysis using *cis*-eQTLs calculated on 'pseudo-bulked' expression across all cell types revealed that less than half of the colocalizations would have been detected in an equivalently sized bulk-tissue dataset (Fig. 3b). Conversely, a small proportion (12.9%) of colocalization was only detected when combining expression signals across all cell types.

The largest number of colocalizations were for schizophrenia ($n_{coloc}$ = 86) and intelligence ($n_{coloc}$ = 79), followed by AD ($n_{coloc}$ = 35). Several colocalizations were shared across phenotypic categories, suggesting potential shared etiology (Supplementary Fig. 10). In Fig. 3c–e, we show the cell-type-specific colocalization probabilities (PP.H4 > 0.8) for AD, highlighting several genes known to confer risk to AD such as *BIN1* (PP.H4$_{microglia}$ = 1.0), *RASGEF1C* (PP.H4$_{microglia}$ = 1.0) and *PICALM* (PP.H4$_{microglia}$ = 1.0), as well as less well-established AD risk genes such as *SNX31* (PP.H4$_{astrocytes}$ = 0.99), *JAZF1* (PP.H4$_{microglia}$ = 0.98; Fig. 3d) and *EGFR* (PP.H4$_{astrocytes}$ = 1.0; Fig. 3e).

We then investigated the impact of disease on colocalization using the interaction framework described above based on the lead colocalized SNP proposed by COLOC. Because this assessment was conducted on a small number of genes in each cell type, we considered nominal LRT *P* value < 0.05 indicative of an interaction between disease diagnosis and allelic regulation of gene expression. Across the full set of 501 colocalizations, 118 (23.6%) showed an interaction between their lead colocalized SNP and disease status (Fig. 4a and Supplementary Table 5). For example, for the colocalized triplet *TP53INP1*–oligodendrocytes–AD (PP.H4 = 0.91), the lead COLOC SNP rs4582532 showed a significant preference for the interaction model (LRT, *P* = 1.45 × 10$^{-2}$), influenced by PD samples (PD$_{interaction}$, *P* = 8.83 × 10$^{-3}$; Figs. 4c,d). Similarly, the lead colocalized SNP rs1691364 for the triplet *RAB38*–excitatory–neurons–FTD (PP.H4 = 0.81) showed a preference for the interaction model (LRT, *P* = 1.10 × 10$^{-3}$) with interaction effects from PD and AD samples (PD$_{interaction}$, *P* = 1.56 × 10$^{-3}$; AD$_{interaction}$, *P* = 3.49 × 10$^{-2}$; Figs. 4e,f).

## MR

The above-mentioned analysis highlights the presence of dynamic disease-dependent effects on genetic regulation of gene expression

in the human brain. In contrast, a principled implementation of eQTL-anchored MR requires gene expression profiles that are unconfounded by disease status. We, therefore, restricted our MR analysis to the subset of 183 control participants with no clinical history of brain disease and no evidence of brain disease on neuropathological examination. Using only control samples, we repeated the single-cell-type *cis*-eQTL discovery using the same methodology as for the full dataset based on residual expression adjusted for clinical covariates (age at death, sex, PMI and sample source as fixed effects) and optimized PCs as fixed covariates for each cell type. We identified 10,470 eGenes (FDR < 5%) across the eight cell types, representing 7,204 unique eGenes (Supplementary Fig. 11 and Supplementary Table 6). As previously, most eGenes (5,046, 70.0%) were observed in only a single cell type.

Despite normal neuropathology and no history of neurological disease, we considered the possibility that occult (premanifest) brain disease might still be present in control samples, particularly in samples from aged participants. To explore potential age-related allelic effects on gene expression, we applied the interaction methodology described above and selected age as the covariate that interacts with the genotype. Taking the lead *cis*-eQTL SNP for each eGene, 1,605 (15.3%) were significantly better modeled with age–genotype as an interaction term (*q* < 0.05). This proportion varied greatly by cell type, ranging from 7.9% (292 of 3,698) in excitatory neurons to 45.1% (83 of 184) in pericytes (Supplementary Table 7). Across all age-interaction eQTLs, 10.8–29.4% (depending on cell type) overlapped with disease–interaction eQTLs, and these loci were therefore excluded from the downstream MR analysis.

To select appropriate instruments for MR, we first repeated the colocalization analysis using control-only eQTLs under the single causal variant hypothesis (PP.H4) and using the same 41 CNS phenotypes assessed above. In total, we identified 256 colocalizations (PP.H4 > 0.8) across 26 CNS phenotypes (Supplementary Table 8), of which 91 (35.5%) were not present in the larger, higher-powered mixed disease-case and control eQTL dataset (Fig. 4h and Supplementary Fig. 12). For example, the colocalization triplet *PEX13*–excitatory–neurons–MS was only present in the control cohort (PP.H4 = 0.87) despite a sample size less than half the full dataset (PP.H4$_{fulldataset}$ = 0.08; Fig. 4g). Analysis of the lead SNP for this colocalization, rs1177284, revealed a significant interaction with all disease categories (LRT, *P* = 3.66 × 10$^{-3}$; AD$_{interaction}$, *P* = 4.08 × 10$^{-2}$; PD$_{interaction}$, *P* = 1.83 × 10$^{-2}$;

**Fig. 3 | Colocalization analysis for brain phenotypes using cell-type eQTLs. a**, Summary of colocalizations (PP.H4 > 0.8) per cell type and trait. Each bar chart is colored by cell type. *Y* axes indicate the number of colocalizations in that cell type. Asterisks indicate the cell type with most colocalizations with a particular trait. **b**, Number of unique colocalized (PP.H4) genes from single-cell-type and 'pseudobulk' eQTL data. **c**, Single cell-type colocalizations (PP.H4 > 0.8) for AD (genes on *x* axis and cell types on *y* axis). **d**, Colocalization Manhattan plots for the association of *JAZF1* expression in microglia and AD risk. Each point represents the −log$_{10}$(*P*) for an SNP and its association with gene expression (top) and disease risk (bottom). **e**, Colocalization Manhattan plots for the association of *EGFR* expression in astrocytes and AD risk. ADHD, attention

deficit hyperactivity disorder; ALS, amyotrophic lateral sclerosis; AN, anorexia nervosa; AUDIT, alcohol use disorder; INT, intelligence; SCZ, schizophrenia; BIP, bipolar disorder; HL, hearing loss; CBV, cerebellar volume; CSA, cortical surface area; FTD, frontotemporal dementia (behavioral variant); HV L/R, hippocampal volume (left/right); ICV, intracranial volume; INS, insomnia; LAN, reading and language skills; LBD, Lewy body dementia; MCP, multisite chronic pain; MDD, major depressive disorder; NDD, neurodegenerative disease; NEUR, neuroticism; PVS, perivascular space burden; RLS, restless legs syndrome; SCV, subcortical volume; SD, sleep duration; STR, stroke; THV, whole thalamus volume; chr, chromosome.

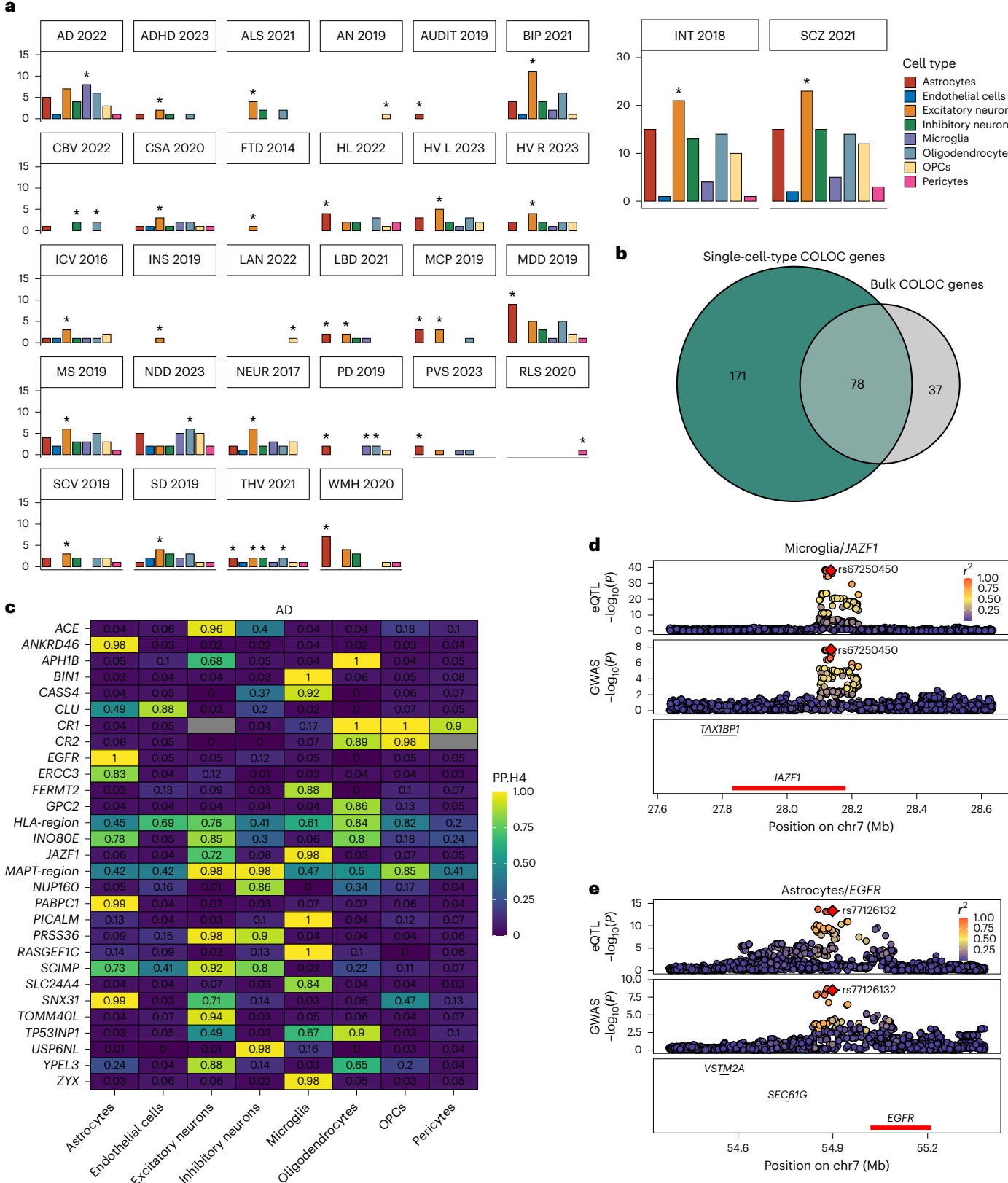

MS$_{interaction}$, $P = 4.51 \times 10^{-2}$; Fig. 4h). This highlights the value of control-only human brain gene expression data in enabling enhanced interpretation of disease-associated variants.

Following colocalization analysis, we performed linkage disequilibrium pruning ($r^2 < 0.01$) based on the lead SNP from COLOC and retained independent SNP–transcript associations as instrumental

variables (IVs; >90% of exposures retained a single IV). Before MR, we assessed the robustness of the selected IVs using the $F$ statistic[15] and retained IVs with an $F$ statistic >15 (Supplementary Fig. 13). Excluding the MAPT and HLA loci, we found significant MR evidence for an association between genetically proxied transcript levels in a specific cell type and a trait for 94 unique genes across 22 diverse CNS phenotypes

(140 gene/cell type/trait triplets; Fig. 5a and Supplementary Table 9). The majority (81.9%) of genes inferred to have a putative causal association with a CNS trait were found in a single cell type. For 13 genes, there was evidence that a change in expression was associated with more than one CNS phenotype (Supplementary Fig. 14).

While careful selection of IVs, as described above, minimizes the risk of spurious MR findings, the application of single-instrument MR is sensitive to the particular choice of variants and ignores the majority of genetic data in the colocalized region. As a technical validation of these results, therefore, we implemented the multi-instrument MR method principal components analysis–inverse variance weighting (PCA-IVW)[16], which takes account of the full set of variants in the colocalized region. Of the 140 significant MR hits, 138 (98.6%) were replicated using PCA–IVW with the same direction of effect (Supplementary Table 9).

In addition to inferring associations between genes, cell types and outcomes, MR in control participants informs the directionality of the relationships unconfounded by disease-induced changes in gene expression. Knowledge of the directionality of a putatively causal relationship is critical to informing the therapeutic strategy (target activation or inactivation). For example, among the genes with MR evidence of association with AD (Fig. 5b), genetically predicted increased *EGFR* expression is associated with increased AD risk (MR$-P_{IVW} = 2.35 \times 10^{-9}$, $\beta_{IVW} = 0.14$), supporting the potential for repurposing brain-penetrant EGFR inhibitors to the treatment of AD[17]. Similarly, we observed an association between genetically predicted increased expression of *GPNMB* and PD in astrocytes (MR$-P_{IVW} = 1.02 \times 10^{-8}$, $\beta_{IVW} = 0.27$) and OPCs ($P_{IVW} = 1.47 \times 10^{-8}$, $\beta_{IVW} = 0.15$), and this directionality of effect between GPNMB and PD was recently supported by the experimental demonstration that loss of GPNMB activity reduces cellular internalization of fibrillar α-synuclein and related pathogenicity[18].

**Evaluation of single-cell MR inferences in the human proteome**
Because associations between genetically proxied transcripts and outcomes are expected to be mediated by a gene's protein product, we assessed the extent to which the association of a CNS outcome with an exposure converges across both transcript and protein levels. Here we first used data on circulating blood protein levels from the UK Biobank Pharma Proteomics Project (UKB-PPP)[19], which assayed 2,923 plasma proteins in 54,219 participants. Of the 94 genes with single-cell MR evidence to support a causal association with one or more CNS phenotypes, 15 (16.0%) had a measurable protein in plasma, representing 20 gene–trait pairs. For each pair, we re-assessed the evidence for a single causal variant between the protein's expression in plasma and its paired CNS outcome, finding replicable colocalization for 16 (80%) of the gene–trait pairs at PP.H4 > 0.8. Exposure IVs were selected as described above, with median of 7 independent IVs per plasma protein (range = 1–9). Using two-sample MR, 11 of 20 gene–trait pairs (55%) had a significant association between the change in the level of the gene's protein product in plasma and the same CNS outcome identified by single-cell MR (Fig. 5c,d, Supplementary Table 10 and Supplementary Fig. 15). Examples of proteins with a significant association between a change in their level in plasma and a CNS phenotype that were also putatively causal in single-cell MR include CR1-AD (IVW$_{plasma protein}-$ $P = 2.10 \times 10^{-2}$, $\beta = 0.11$), GPNMB-PD (IVW$_{plasma protein}-P = 1.80 \times 10^{-7}$, $\beta = 0.34$) and TNFRSF1A-MS (IVW$_{plasma protein}-P = 1.02 \times 10^{-13}$, $\beta = 2.14$). The independent replication of these and other (Fig. 5c,d) single-cell MR inferences using plasma proteins highlights them not just as candidate drug targets for disease modification but as candidate peripheral biomarkers for use as intermediate phenotypes predictive of clinical outcomes following therapeutic intervention.

Given the sparsity of proteins assayed by UKB-PPP, we attempted to validate genes associated with CNS outcomes by single-cell MR using a second proteomic resource derived from the brains of 330 older adults[20]. Although of limited replication value due to the inclusion of disease cases (31% of participants had a diagnosis of AD) and the unavailability of genome-wide SNP data precluding independent colocalization analysis, of the 94 single-cell MR genes identified in our study, 44 (46.8%) had a measurable protein level (50 gene–trait pairs). From these, we selected proteins with significant (FDR < 5%) protein QTLs (pQTLs) and retained independent IVs ($r^2 < 0.01$) for 21 proteins representing 22 gene–trait pairs. Of these 22 pairs, 20 (90.9%) had a significant (IVW $P < 0.05$) association to the same trait as predicted by single-cell MR, including GPNMB-PD (IVW$_{brain protein}-P = 2.48 \times 10^{-8}$, $\beta = 0.39$), SCFD1-amyotrophic lateral sclerosis (IVW$_{brain protein}-$ $P = 1.14 \times 10^{-13}$, $\beta = 1.61$) and ICA1L-WMH (IVW$_{brain protein}-P = 2.51 \times 10^{-13}$, $\beta = 1.27$; Fig. 5e, Supplementary Table 11 and Supplementary Fig. 16).

## Discussion
MR analysis using molecular phenotypes offers an approach to prioritize drug targets, inform the cell types in which they act and identify biomarkers predictive of clinical outcomes. A key finding from our study was the extent to which brain disease impacts the relationship between genetic variation and gene expression in individual cell types and the implications of this for interpreting eQTL pathogenicity. Notably, we show that disease-dependent allelic effects on gene expression are not fully accounted for by adjusting gene expression for disease status, which has been the standard approach to date (for example, refs. 7,21). In contrast, a principled implementation of MR requires genetic variants instrumenting proximal molecular exposures that predate the onset of disease. To identify the effect of genes in single cell types unconfounded by brain disease, we undertook a principled analysis using only nondiseased control human brain samples. Across 26 diverse CNS phenotypes, we identified 256 gene–cell-type–trait triplets supported by colocalization, of which 35.5% were not detected in the larger, better-powered mixed disease-case and control dataset. This highlights the value of control-only samples in the interpretation of eQTL pathogenicity, as well as the potential for putatively causative associations to be obscured when mixing disease-case and control samples.

**Fig. 4 | Impact of disease on colocalizations. a**, Overview of colocalizations (PP.H4 > 0.8) aggregated across cell types for 30 CNS traits, indicating the total number of colocalizations for a given trait and the proportion with a disease interaction. **b**, Colocalization between *TP53INP1* expression in oligodendrocytes and AD in the full cohort (left, PP.H4 = 0.91) and the control-only cohort (right, PP.H4 = 0.04). Each point represents the $-\log_{10}(P)$ for an SNP and its association with the gene (top) and disease (bottom). **c**, *Cis*-eQTL plot showing the effect of disease samples in the full cohort ($n = 391$) on the association between the lead colocalized SNP rs4582532 and *TP53INP1* expression in oligodendrocytes. The $P$ values for 'all' represent the $t$ statistic for the $M_1$ models, whereas the $P$ values for AD, PD and MS represent the $P$ value from the interaction with genotype within the $M_2$ model. The 'control' $P$ value represents the effect of genotype on expression after accounting for interaction effects. **d**, Colocalization between *RAB38* expression in excitatory neurons and

behavioral frontotemporal dementia in the full cohort (left, PP.H4 = 0.81) and the control-only cohort (right, PP.H4 = 0.06). **e**, *Cis*-eQTL plot showing the effect of disease samples in the full cohort ($n = 391$) on the association between the colocalized lead SNP rs16913634 and *RAB38* expression in excitatory neurons. Elements of the boxes show the center line (median), box limits represent upper and lower quartiles and whiskers represent upper and lower quartiles ±1.5× IQR. All data points have been included. **f**, Colocalization between *PEX13* expression in excitatory neurons and MS in the full cohort (left, PP.H4 = 0.08) and the control-only cohort (right, PP.H4 = 0.87). **g**, *Cis*-eQTL plot showing the effect of disease samples in the full cohort ($n = 391$) on the association between the colocalized lead SNP rs11772842 and *PEX13* expression in excitatory neurons. **h**, Comparison of colocalizations discovered in the full mixed disease-case and control dataset ($n = 391$) versus control samples only ($n = 183$).

Methodologically, we used colocalization to identify a single common genetic region in a particular cell type before IV selection for MR. As noted by others, the inclusion of *cis*-regulated instruments identified by prior colocalization has the advantage of limiting the likelihood of confounding by horizontal pleiotropy, and target-indication pairs selected based on combined evidence from both colocalization and MR have a higher likelihood of regulatory approval after clinical trials[22]. Following colocalization, we selected independent variants at

the colocalized locus as genetic instruments. Although this avoids inflating the MR results from multiple correlated instruments, the number of independent instruments for gene exposure in a single cell type was often small, limiting our ability to effectively control for false discovery[23].

A key question arising from our observation of disease effects on allelic associations with gene expression is whether gene–trait associations informed by genetic regulation of gene expression in diseased

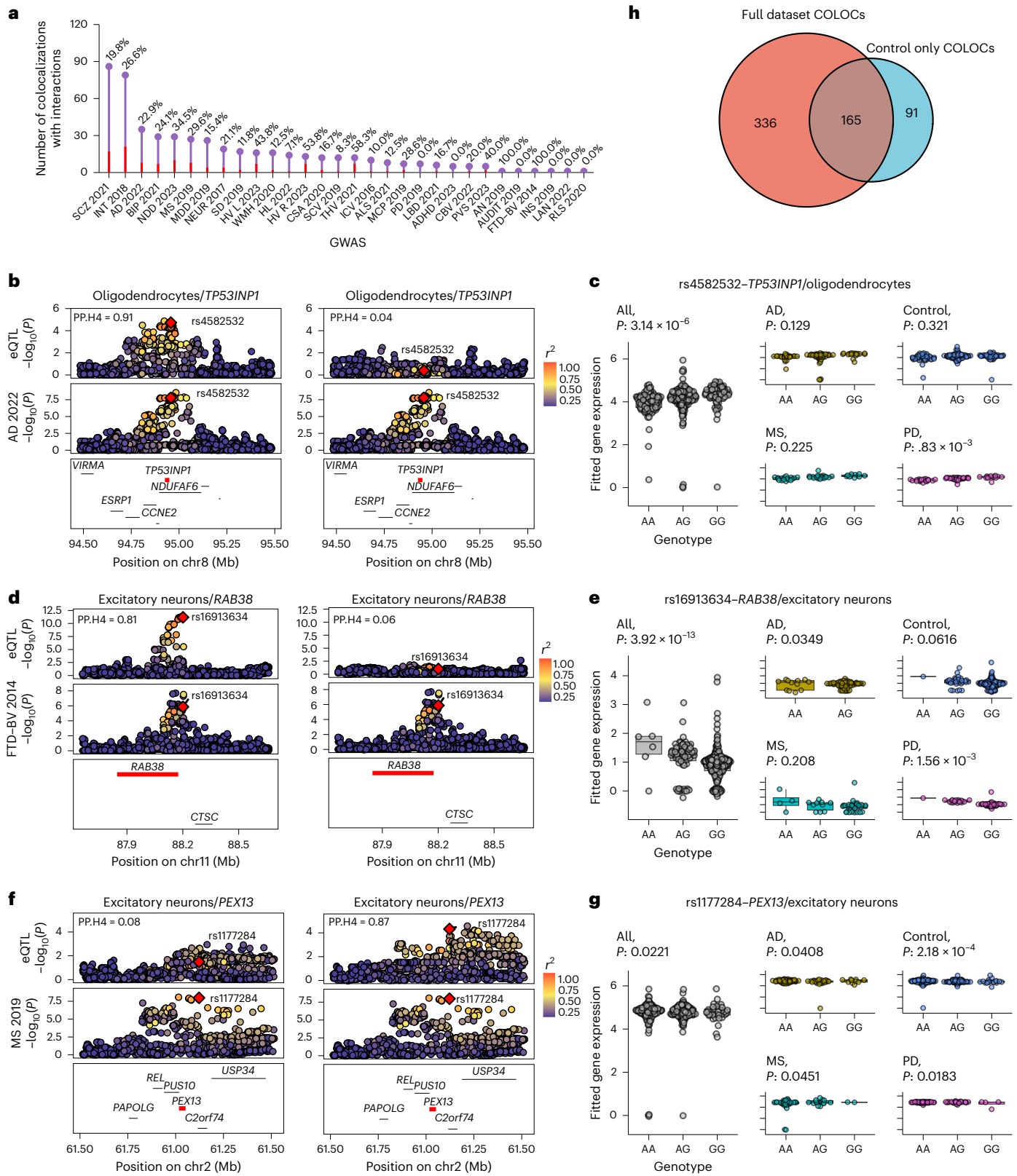

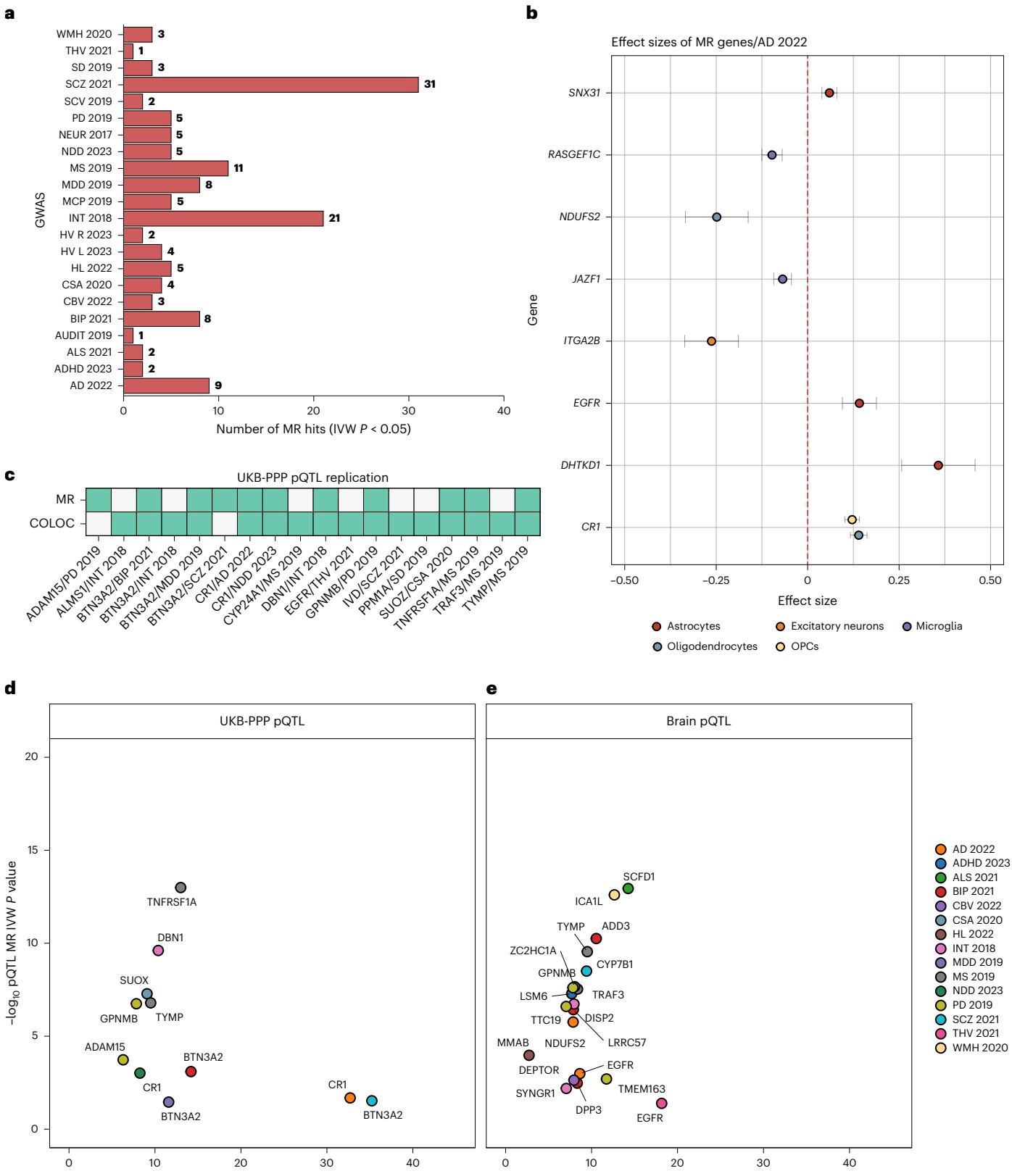

**Fig. 5 | MR results for brain phenotypes at gene and protein expression levels.**
**a**, Overview of significant MR results (IVW fixed effects *P* < 0.05 or Wald ratio
for single instruments). Trait abbreviations are as per Fig. 3b. **b**, IVW effect
size and directionality for genes with MR evidence for an association between
the change in gene expression in the indicated cell type and risk of AD.
Each point represents the IVW effect size for a given cell type, with error bars
(effect size ± 1.96× s.e.) indicating the 95% confidence interval. A positive effect
size means an increase in expression is associated with an increase in disease risk

(and vice versa), whereas a negative effect size indicates an inverse association
between gene expression and disease risk. **c**, Overview of single-cell gene–trait
pairs replicated by colocalization analysis and/or MR using plasma pQTLs
derived from the UKB-PPP. **d**, Overview of MR IVW *P* value associations for pQTLs
in plasma compared to the corresponding single-cell MR IVW *P* value (*x* axis) at
−log₁₀ scale. **e**, Overview of MR IVW *P* value associations for pQTLs in brain (*y* axis)
compared to corresponding single-cell MR IVW *P* value (*x* axis) at −log₁₀ scale.

brain samples represent plausible causative associations. A causative interpretation would require either that the genetic regulation of the exposure in a cell type is unaltered between disease and control samples or that the effects of genetic variation are exerted only in the presence of pathology. In the latter scenario, risk alleles could be conceived as an 'Achilles' heel'[24], accelerating disease progression in the presence of pathology such that they manifest as disease susceptibility alleles in genetic association studies. As such, we propose that the study of both control and disease samples is complementary in terms of identifying plausible drug targets for risk mitigation and disease treatment. However, because treatment targets may be repurposed for risk mitigation and vice versa, it is incumbent on researchers to clarify the relationships between exposures and outcomes in both diseased and nondiseased brains.

Although the value of single-cell gene expression data is well-established, we highlight that of the hundreds of colocalizations reported here, less than half would have been detected using *cis*-eQTLs derived from an equivalently sized bulk-tissue gene expression dataset. Conversely, we noted that a small proportion of eGenes were only detected when combining gene expression signals across all cell types, suggesting leveraging shared genetic regulation across cell types may provide additional useful gains in power to detect some SNP–transcript associations, albeit at the expense of information about cell-type specificity.

In situations where multiple cell types are implicated in a particular target-indication pairing, our analytical framework is unable to determine if this reflects a causal mechanism manifesting across multiple cell types or simply shared genetic regulation across different cell types. A further limitation is the lack of available large GWAS for many brain phenotypes. In this study, we only included genome-wide significant GWAS loci and implemented a strict PP.H4 cut-off of 0.8 or higher. In some circumstances, exploratory discovery analyses at more liberal GWAS *P* values and lower PP.H4 cut-offs may be an appropriate first step.

In contrast to the limited power of current single-cell brain datasets, the UKB-PPP is a well-powered proteomic dataset for pQTL mapping in human plasma. Of the 140 gene–trait associations identified by single-cell MR in control participants, only 20 included a target protein measurable in plasma, but of these, we observed high rates of replication (80% for colocalization and 55% by MR). Given the substantial power of UKB-PPP to detect multiple independent IVs for these proteins (median of 7 per candidate target), UKB-PPP provides a robust and independent replication of the single-cell-type MR inferences for these targets. The high level of replication in plasma also supports the principle that assay of circulating proteins identified by MR in the brain may also have value as peripheral biomarkers predictive of clinical outcome and therapeutic response.

In addition to providing a principled framework for assessing causal relationships, our implementation of MR provides information on the directionality of the association between exposure and outcome in a specific cell type. Knowledge of the direction of effect is necessary to guide the directionality of the therapeutic intervention, while knowledge of the cell types via which genes act can aid the design of more precise preclinical experiments, including the assessment of target engagement in relevant tissues. The identification of shared risk factors across disease categories presents opportunities for shared preventative strategies, drug repurposing and prediction of adverse events[25].

In conclusion, this study reports a generalizable framework for the principled conduct of MR in single cell types. The study advances our understanding of the influence of disease on eQTLs in the human brain and highlights the importance of considering disease effects when assessing eQTL pathogenicity. We prioritize candidate targets for brain disease in their cellular context, and by using healthy control brain tissue, we establish a resource for the unbiased and enhanced interpretation of GWAS risk alleles for human brain disease, structure and behavior.

## Online content

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

## Methods

### Ethics statement

We complied with all relevant statutory and ethical regulations approved by the Imperial College Research Ethics Committee regarding the use of human postmortem tissue.

### Dataset summary

snRNA-seq and genotype data were generated on 409 deceased individuals of European ancestry from four datasets. These include 'MRC_60' (new samples generated in-house, 60 control individuals), 'Roche_PD' (new samples generated in-house, 92 individuals, including 70 PD cases and 22 controls), 'Bryois_192' (192 total, including 105 mixed disease and 87 control cases obtained from their recent publication[7]) and 'MATTHEWS' (new samples generated in-house consisting of 65 individuals split into 38 AD cases and 27 controls), a subset of which have been used in published work[26,27]. Following quality control (QC) and data harmonization, we retained samples from a total of 391 individuals (183 control brains and 208 diseased brains).

### Samples

Ethics approval for the work carried out on postmortem human brain tissue was given by the Wales REC3 Ethics Committee (REC reference 18/WA/0238). Study analyses complied with the Imperial College Research Ethics Committee (ICREC reference 14/02/11). At the individual brain banks, postmortem, fresh tissue samples were snap-frozen in liquid nitrogen vapor for 20 min before being stored in a −80 °C freezer for the long term. Immunohistochemistry was undertaken on all samples using adjacent brain tissue (same block) and assessed for β-amyloid, Tau, TDP43, α-synuclein and p62. All hematoxylin and eosin stains were performed by hand. In the selection of control samples, we excluded all samples with a premortem history of neurological or psychiatric disease (at any time), and in all cases, there was no evidence of neurodegenerative or other significant diseases on neuropathological examination.

### Nuclei isolation and snRNA-seq

For the MRC_60 dataset, snRNA-seq data were generated at the Imperial College on the prefrontal cortex and hippocampus samples ascertained from 60 unrelated participants ascertained from the Imperial College, Oxford University, Edinburgh University or Amsterdam Medical Center brain tissue banks. Nuclei were isolated as previously described[28], with a slightly extended douncing during tissue lysis (see ref. 29 for detailed protocol). Then, snRNA-seq data were generated using the 10x Single Cell Next GEM Chip targeting a minimum of 5,000 nuclei per sample and libraries prepared using the Chromium Single Cell 3′ Library and Gel Bead v3 kit according to the manufacturer's instructions. cDNA libraries were sequenced using the Illumina NovaSeq 6000 system at a minimum sequencing depth of 30,000 paired-end reads per nucleus.

For the Roche_PD dataset, nuclei were isolated using the Nuclei Pure Prep Nuclei Isolation Kit (Sigma-Aldrich) with the following modifications. The tissue was lysed in nuclei pure lysis solution with 0.1% Triton X-100, 1 mM dithiothreitol (DTT) and 0.4 U μl⁻¹ SUPERaseIn RNase Inhibitor (Thermo Fisher Scientific) freshly added before use and homogenized with the help of first a 23G and then a 29G syringe. Cold 1.8 M sucrose cushion solution, prepared immediately before use with the addition of 1 mM DTT and 0.4 U μl⁻¹ RNase Inhibitor, was added to the suspensions before they were filtered through a 30 μm strainer. The lysates were then carefully and slowly layered on top of 1.8 M sucrose cushion solution previously added into new Eppendorf tubes. Samples were centrifuged for 45 min at 16,000$g$ at 4 °C. Pellets were resuspended in nuclei storage buffer with RNase inhibitor, transferred in new Eppendorf tubes and centrifuged twice for 5 min at 500$g$ at 4 °C. Finally, purified nuclei were resuspended in nuclei storage buffer with RNase inhibitor, stained with trypan blue and counted using Countess II (Life Technology). After the count, nuclei permeabilization

was carried out following the demonstrated protocol for single-cell multiome ATAC + Gene Expression sequencing from 10× Genomics. A total of 12,000 estimated nuclei from each sample were used for the transposition step and then loaded on the Chromium Next GEM Single Cell Chip J. ATAC library and gene expression library construction was performed using the Chromium Next GEM Single Cell Multiome ATAC + Gene Expression kit according to the manufacturer's instructions. Libraries were sequenced using the Illumina NovaSeq 6000 System and the NovaSeq 6000 S2 Reagent Kit v1.5 (100 cycles), aiming at a minimum sequencing depth of 30k reads per nucleus.

For the MATTHEWS dataset, homologous fresh frozen brain tissue blocks from the entorhinal cortex, middle temporal gyrus and somatosensory cortex were cryosectioned at 80 μm, 200 mg of gray matter was collected in RNAse-free Eppendorf tube and nuclei were isolated as previously described[26]. All steps were carried out on ice or at 4 °C. Tissue was homogenized in buffer (1% Triton X-100, 0.4 U μl⁻¹ RNAseIn + 0.2 U μl⁻¹ SUPERaseIn, 1 μl (1 mg ml⁻¹) DAPI) using a 2 ml glass douncer. The homogenate was centrifuged at 4 °C for 8 min at 500$g$, and the supernatant was removed. The pellet then was resuspended in a homogenization buffer and filtered through a 70 μm filter followed by density gradient centrifugation at 13,000$g$ for 40 min. The supernatant was removed, and nuclei were washed and filtered in PBS buffer (PBS + 0.5 mg ml⁻¹ BSA + 0.4 U μl⁻¹ RNAseIn + 0.2 U μl⁻¹ SUPERaseIn). Nuclei were pelleted, washed twice in PBS buffer and resuspended in 1 ml PBS buffer. In total, 100 μl of nuclei solution was set aside on ice for single nuclear processing. Isolated nuclei stained with acridine orange dye were counted on a LUNA-FL Dual Fluorescence Cell Counter (Logos Biosystems, L20001). Approximately 7,000 nuclei were used for 10× Genomics Chromium Single Cell 3′ processing and library generation. All steps were conducted according to the 10x Genomics Chromium Single Cell 3′ Reagent Kits v3 User Guide, with eight cycles of cDNA amplification until fragmentation, where 25 ng of amplified cDNA per sample was taken through for fragmentation. The final index PCR was conducted at 14 cycles. cDNA and library prep concentrations were measured using the Qubit dsDNA HS Assay Kit (Thermo Fisher Scientific, Q32851), and DNA and library preparations were assessed using the Bioanalyzer High-Sensitivity DNA Kit (Agilent Technologies, 5067-4627). Pooled samples at equimolar concentrations were sequenced on an Illumina HiSeq 4000 according to the standard 10x Genomics protocol.

### Genotyping

For the MRC_60 dataset, donor DNA from samples processed at the Imperial College were genotyped using the Illumina Infinium Global Screening Array (v2.0). The Roche_PD dataset was genotyped with the same method described previously, as well as the Bryois_192 dataset (which includes whole-genome sequencing)[7]. For the MATTHEWS dataset, DNA was extracted from human postmortem tissue using the DNeasy kit (Qiagen) with the recommended protocol. Briefly, samples were lysed overnight with proteinase K. Lysate was loaded into DNeasy Mini spin columns with supplied buffers for centrifugation rounds before the elution of DNA. DNA concentration was determined using a NanoDrop (Thermo Fisher Scientific) and Qubit assay (Thermo Fisher Scientific). Genotyping was performed at Cambridge Genomic Services at the University of Cambridge with the UK Biobank Axiom Array. All genotypes then underwent imputation on the Michigan Imputation Server (v.1.6.3) using the Haplotype Reference Consortium (v.r1.1) reference panel of the European population[30,31] with a prephasing using Eagle (v.2.4)[32] and imputation using Minimac4 (ref. 31).

Genotype data from each vcf underwent a series of quality control steps using PLINK (v.2.0)[33]. SNPs with an imputation score <0.4 were removed from the analysis, as well as SNPs with missingness >5%. Multi-allelic (>2 alleles) SNPs were also excluded from the analysis, as well as SNPs deviating from Hardy–Weinberg equilibrium ($P < 1 \times 10^{-6}$). We also restricted SNPs in autosomal chromosomes 1–22 and removed

any individuals with more than 2% missing genotypes. Before merging, each vcf underwent a final check using bcftools (v.1.18)[34] against the latest ENSEMBL hg38 genome build fasta file[35], flipping alleles and genotype calls for mismatches. Following merging, we excluded all individuals with kinship >0.2 (indicating duplicated individuals). Finally, we retained 5.20 million high-quality SNPs in 391 individuals.

## Single-cell analysis

Both the raw sequencing files from the MATTHEWS datasets and Bryois_192 were mapped using CellRanger to the GRCh38 reference genome as previously described[7,26]. For the MRC_60 dataset, the raw sequencing reads were mapped to the GRCh38 genome and quantified gene counts as unique molecular identifiers (UMIs) using Cell Ranger count (v.5.0.1). We counted reads mapping to introns as well as exons by --include-introns option in Cell Ranger (v.5.0.1). As shown in the earlier studies, this results in a greater number of genes detected per nucleus, as well as better cell-type classification[36,37]. Finally, the reference genome was created using Cell Ranger mkref (v.5.0.1) with default settings[38]. In addition, sample pools were demultiplexed based on their genotype using the Demuxlet algorithm with the default settings, as previously described[39,40]. The variable SNPs between the pooled individuals were used to determine which cell belongs to which individual and to identify doublets. Droplets called doublet by Demuxlet were removed from downstream analyses. For the Roche_PD dataset, raw FASTQs were aligned to the GRCh38 genome using CellRanger-ARC count (v.2.0.2). After mapping, RNA counts were extracted to construct Seurat objects.

On the MATTHEWS, MRC_60 and Roche_PD datasets, we also assessed the proportion of empty droplets using the EmptyDrops tool (from DropletUtils v.1.22) package, retaining nuclei with an FDR < 0.01 (ref. 41). Seurat objects were generated using Seurat (v4)[42], retaining nuclei with at least 500 UMIs in 300 features and with less than 5% mitochondrial content. We further identified doublets using DoubletFinder (v.2.0), which were then removed[43]. Finally, all samples within each dataset were integrated using reciprocal PCA within Seurat. The final QC'd merged Seurat object within each dataset was then used to assign cell types using canonical markers, specifically excitatory neurons (SLC17A7 and SATB2), inhibitory neurons (GAD1 and GAD2), astrocytes (AQP4 and FGFR3), microglia (C1QB and CSF1R), OPC (PDG-FRA and VCAN), oligodendrocytes (MAG and MOG), pericytes (RGS5) and endothelial cells (CLDN5). Finally, we performed ambient RNA removal using DecontX[44] using default parameters and cell-type labels and obtained corrected counts for each cell in all datasets (including Bryois_192).

## eQTL mapping

The ambient RNA-corrected count matrices were extracted from each annotated cell type, after which the counts for all cells were summed per individual to obtain a single aggregated count value per cell type per individual (pseudobulking). Only individuals with at least ten cells in a specific cell type were included. Following aggregation, we normalized count matrices using a log transformation on the counts per million values obtained using the edgeR package (v.3.42)[45]. To increase the comparability across transcripts, we scaled the expression matrices. Consequently, the interpretation of the expression values is that each unit represents a change in one s.d. Only genes expressed in at least 5% of individuals were kept for further analysis.

For both the controls-only and full datasets, the mapping of *cis*-eQTLs was performed in two steps. We first fitted a linear model for the expression of each gene against clinical and technical covariates, which included age, sex, PMI, sample source and diagnosis as fixed effects, from which we obtained the residual gene expression. *Cis*-eQTL discovery was performed using MatrixEQTL (v.2.3)[9] and a *cis*-distance of 1 Mb as input to the *cisDist* parameter, including all technical and clinical covariates. We included genotype PCs to account for potential

population structure and tested the optimal number of expression PCs on the residual expression, specifically for each cell type to be added as linear covariates for a final model:

$$E \sim G + \text{expression PCs}\,(1-n) + \text{genotype PCs}\,(1-5),$$

where $E$ is the residual gene expression corrected for covariates, $G$ is the genotype dosage for a specific SNP and PCs $(1-n)$ are dependent on the maximum eGenes discovered for $n$ PCs in a specific cell type. We then filtered the MatrixEQTL outputs by FDR using the Benjamini–Hochberg method[46] for both sets of eQTL discoveries.

## Interaction modeling

From the eQTL discovery on the full dataset (controls + disease cases), we obtained all the eQTLs for each cell type below 5% FDR. Using the top SNP for each gene, we re-assessed the reported associations using an LME model with the use of the lmerTest package (v.3.1)[47].

We first tested an LME model that can be categorized as the null model ($M_0$), modeling the gene versus all covariates:

$$E \sim 1 + \text{diagnosis} + \text{age} + \text{PMI} + \text{sex} + (1 + \text{diagnosis}|\text{sample source})$$
$$+ \text{expression PCs}(1-n) + \text{genotype PCs}(1-5).$$

The replication model ($M_1$), which includes the addition of genotype $G$ to replicate the MatrixEQTL model:

$$E \sim 1 + G + \text{diagnosis} + \text{age} + \text{PMI} + \text{sex} + (1 + \text{diagnosis}|\text{sample source})$$
$$+ \text{expression PCs}\,(1-n) + \text{genotype PCs}\,(1-5).$$

Finally, the interaction model ($M_2$) to test for interactions between diagnosis $D$ and genotype $G$:

$$E \sim 1 + G \times \text{diagnosis} + \text{age} + \text{PMI} + \text{sex} + (1 + \text{diagnosis}|\text{sample source})$$
$$+ \text{expression PCs}\,(1-n) + \text{genotype PCs}\,(1-5).$$

For each model, we performed an LRT implemented in the package lmerTest (v3.1)[47] between the baseline and the more complex model ($M_0$ versus $M_1$, and if significant, $M_1$ versus $M_2$)[48] to assess which model better fits a particular SNP–gene association. In the LRT, a significant test ($P < 0.05$) suggests that a complex model is statistically more appropriate.

For the models using age as an interaction term, we restricted the analyses to the control-only dataset for simplicity, as it may not be possible to disentangle the combined effect of age and the presence of disease. In this scenario, we used the following model:

$$E \sim 1 + G * \text{age} + \text{PMI} + \text{sex} + (1|\text{sample source})$$
$$+ \text{expression PCs}\,(1-n) + \text{genotype PCs}\,(1-5).$$

Finally, we performed multiple-testing correction on the LRT $P$ values by using the $q$ value package (v.2.34)[49] and labeled each association with an LRT $q$ value < 0.05 as better modeled with an interaction term, as implemented previously[11].

## Genetic colocalization

We first processed summary statistics from each GWAS study using the format_sumstats function from the MungeSumstats package (v.1.10.1)[50] to harmonize column names. In cases where effect sizes and s.e. were missing, we enforced the imputation of these values using the impute_se = TRUE and impute_beta = TRUE parameters of the format_sumstats function. In the majority of cases, we also lifted over the SNP positions from hg19 to hg38. To select regions for colocalization analysis, we first scanned each GWAS using the ld_clump function of the ieugwasr (v.1.0.1)[51] package to retain the top variant in windows of 1 Mb and avoid overlapping regions. We then repopulated each

window with the full set of SNPs from the GWAS summary statistics to get a full set of regions to test in a colocalization setting. Regions with less than 100 SNPs were excluded from colocalization analysis. In addition, we aggregated genes in the *MAPT* and *HLA* loci in our results by cell type/trait given the potential inaccuracy in individual transcript quantification at these loci[52–54], but report each of these individually in Supplementary Tables 4 and 8.

For each trait, we obtained the MatrixEQTL outputs and intersected the SNPs with the GWAS summary statistics. We then iteratively performed genetic colocalization using the coloc.abf function from the COLOC package (v.5.2.3)[13], each time testing one cell-type/gene combination (such as microglia/*PICALM*) as trait 1 against the GWAS as trait 2. For example, in a single region of 1 Mb containing five genes, we would perform 5 (number of genes) × 8 (number of cell types) = 40 colocalization tests. To estimate sdY, we used available minor allele frequency (MAF) information from each SNP for each study; sdY was set as 1 for eQTL associations given the standardization of expression values before eQTL mapping. In addition, we specified type = 'cc' for case–control traits and type = 'quant' for quantitative traits as well as the *cis*-eQTL set. For downstream analyses, we retained colocalizations with a posterior probability (PP.H4) above 0.8.

## MR

Using the colocalization results of the control-only dataset, we carried out MR on traits (cell type/gene/GWAS) with a colocalization PP.H4 > 0.8. In addition, genes in the *HLA* and *MAPT* loci were excluded completely. Centering the filtering around the proposed lead SNP by the colocalization analysis as the IV, we kept all eQTLs with an association to the gene below 5% FDR and confirmed that the *F* statistic values were all above 15 to ensure that all IVs were robust (Supplementary Fig. 13). To ensure independence between candidate instruments, we further excluded correlated genetic variants ($r^2 > 0.01$). We then applied two-sample MR using the mr_ivw function from the MendelianRandomization package (v.0.10)[55], as more than 90% of traits only retained a single IV, where the analysis is equivalent to the Wald ratio. In all cases, we used the cell-type-specific effect sizes as the exposure and the GWAS effect size as the outcome, where results with IVW *P* value < 0.05 were considered significant.

To further cross-validate these results, we implemented PCA–IVW[16]. Briefly, this method summarizes the genetic associations of all genetic variants within the region into orthogonal PCs. Correlation matrices were obtained using the ld_matrix_local() function from the ieugwasr package, using the 1000 Genomes European reference dataset[51]. We first decomposed the genetic correlation matrix into PCs that explain 99.9% of the variance across the set of genetic variants, which were then used as independent IVs for MR analysis using the IVW method.

## Assessment of MR genes at the protein level

We obtained the full published genome-wide summary statistics for each assessed protein in the brain proteome study detailed in ref. 4, as well as all proteins (2,940) in the European (discovery) cohort as reported by the UKB-PPP[19], and kept summary statistics that intersected with our MR genes. Because the reported summary statistics only included nominal *P* values in the UKB-PPP summary statistics, we performed multiple-testing corrections on the associations using the R (v.4.3.3) p.adjust() function specifying FDR[46]. To independently test a causal association between genetically regulated protein levels and a trait, we applied the same methodology as for the cell-type eQTLs. We took all SNPs within the genome-wide significant regions of each trait (500 kb of each side of the lead GWAS SNP for a total 1 Mb window) and filtered associations <5% FDR. We then removed correlated genetic variants ($r^2 > 0.01$) using the ld_clump() method from the ieugwasr package and then harmonized the effect sizes based on reported effect alleles (A1) in the pQTL summary statistics and GWAS summary statistics

(A2 after formatting with MungeSumstats). We then performed MR using the mr_allmethods() function on the inputs and retained associations with IVW *P* value < 0.05 (ref. 55). In addition, we performed genetic colocalization on the UKB-PPP associations (which supplied genome-wide effect sizes and s.e.) using the same method as the cell-type eQTL colocalizations, specifying the prior probability $p\_12 = 0.01$ given our previous evidence of cell-type-specific colocalization.

## Figures

Most figure panels were generated programmatically in R using the package ggplot2 (ref. 56) or with BioRender (full license) (Fig. 1).

## Statistics and reproducibility

Statistical analyses have been described throughout the manuscript, figure legends and Methods, and performed using R (v.4.3.3), including associated packages (Methods). Version numbers from each package have been included. No statistical analysis was used to predetermine sample size, and excluded samples are described in the Methods.

## Reporting summary

Further information on research design is available in the Nature Portfolio Reporting Summary linked to this article.

## Data availability

Raw snRNA-seq and genotype data from the Bryois_192 dataset are available as per their publication at the European Genome–Phenome Archive (EGA) under accession code EGAS00001006345 (ref. 7). Raw snRNA-seq and genotype data from the MATTHEWS dataset are hosted at Synapse under accession code syn54083444. Newly generated raw snRNA-seq and associated genotype data (MRC_60 and Roche_PD) are available under accession code EGAS50000000687. Genotype data are considered personal data and are therefore under protected access by the host repository (EGA), where access is subject to the submission of an application delineating the scope of the project and the data required (full details on the portal). Applications are aimed to be reviewed within 2 weeks.

Processed single-cell expression counts for each dataset and the full set of eQTL summary statistics for both the full and control-only datasets are available at https://zenodo.org/records/13343729. The full set of published GWAS summary statistics used for the colocalization and MR analysis as well as links to the original publications are described in Supplementary Table 3.

## Code availability

The code used for the analysis presented in this study can be found at https://github.com/johnsonlab-ic/singlecell-MR (ref. 57).

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

## Acknowledgements

This work was supported by the UK Research and Innovation (UKRI) Medical Research Council (MRC; awards MR/S02638X/1 and MR/W029790/1). Brain tissue samples and associated clinical and neuropathological data were supplied by the Parkinson's UK Brain Bank at Imperial, funded by Parkinson's UK, a charity registered in England and Wales (258197) and in Scotland (SC037554); the Oxford Brain Bank, supported by the MRC, Brains for Dementia Research (Alzheimer Society and Alzheimer Research UK), Autistica UK and the National Institute for Health and Care Research (NIHR) Oxford Biomedical Research Center; the Edinburgh Brain Bank supported by the MRC; and the Amsterdam Medical Center Brain Bank. We also acknowledge the support of the Epilepsy Society for A.S. through the Katy Baggott Foundation and for M.T. and M.J. from the Department of Health's NIHR Biomedical Research Centers funding scheme. L.B. received support from the Alan Turing Institute under the UKRI Engineering and Physical Sciences Research Council (EPSRC) (EP/N510129/1) and the Marmaduke Sheild Fund. This research was supported by the NIHR Cambridge Biomedical Research Center (NIHR203312). The views expressed are those of the authors and not necessarily those of the NIHR or the Department of Health and Social Care. A.N., P.M.M., N.F., N.W., A.S., J.J. and V.Z. were supported by the UK Dementia Research Institute, which receives its funding from UK DRI, funded by the UK MRC, Alzheimer's Society and Alzheimer's Research UK. We are grateful for funding support from Epilepsy Society UK (to S.P., R.B., J.M. and S.M.S.). This work would not have been possible without the people and their families who so generously donated brain tissue.

## Author contributions

M.R.J. and L.B. conceptualized the study. A.H., V.Z., M.A., Y.Y., J.H.K., L.W., M.O.J., L.M., R.F., L.L., D.C., D.O., K.M., J.J., L.M., N.F., N.W., A.S., J.J., E.A., D.O.G., R.B., A.S., J.J.A. and L.M. carried out the investigation. A.H., V.Z., L.B. and M.R.J. wrote the manuscript. J.B., D.M., P.K.S., S.M.S., M.T., P.M.M., J.A.-A., A.S., J.J., A.N., J.D.M., L.L., A.C.B. and S.P. reviewed and edited the manuscript.

## Competing interests

The authors declare no competing interests.

## Additional information

**Correspondence and requests for materials** should be addressed to Leonardo Bottolo or Michael R. Johnson.

# Reporting Summary

## Statistics

For all statistical analyses, confirm that the following items are present in the figure legend, table legend, main text, or Methods section.

| n/a | Confirmed | |
|---|---|---|
| ☐ | ☒ | The exact sample size (*n*) for each experimental group/condition, given as a discrete number and unit of measurement |
| ☐ | ☒ | A statement on whether measurements were taken from distinct samples or whether the same sample was measured repeatedly |
| ☐ | ☒ | The statistical test(s) used AND whether they are one- or two-sided<br>*Only common tests should be described solely by name; describe more complex techniques in the Methods section.* |
| ☐ | ☒ | A description of all covariates tested |
| ☐ | ☒ | A description of any assumptions or corrections, such as tests of normality and adjustment for multiple comparisons |
| ☐ | ☒ | A full description of the statistical parameters including central tendency (e.g. means) or other basic estimates (e.g. regression coefficient) AND variation (e.g. standard deviation) or associated estimates of uncertainty (e.g. confidence intervals) |
| ☐ | ☒ | For null hypothesis testing, the test statistic (e.g. *F*, *t*, *r*) with confidence intervals, effect sizes, degrees of freedom and *P* value noted<br>*Give P values as exact values whenever suitable.* |
| ☐ | ☒ | For Bayesian analysis, information on the choice of priors and Markov chain Monte Carlo settings |
| ☐ | ☒ | For hierarchical and complex designs, identification of the appropriate level for tests and full reporting of outcomes |
| ☐ | ☒ | Estimates of effect sizes (e.g. Cohen's *d*, Pearson's *r*), indicating how they were calculated |

*Our web collection on statistics for biologists contains articles on many of the points above.*

## Software and code

Policy information about availability of computer code

| Data collection | No software was used for data collection. |
|---|---|
| Data analysis | Packages or software used in this manuscript include the following; Michigan Imputation Server (version 1.6.3), Eagle (2.4), plink (2.0), bcftools (1.18), Cellranger (5.0.1), Cellranger-ARC (2.0.2), DropletUtils (1.22), Seurat (v4), DoubletFinder (2.0), edgeR (3.4.2), MatrixEQTL (2.3), lmerTest (3.1),qvalue (2.34),MungeSumstats (1.10.1),ieugwasr (1.0.1), coloc (5.2.3), MendelianRandomization (0.10).<br><br>Scripts used for data analysis are available here. https://github.com/johnsonlab-ic/singlecell-MR |

For manuscripts utilizing custom algorithms or software that are central to the research but not yet described in published literature, software must be made available to editors and reviewers. We strongly encourage code deposition in a community repository (e.g. GitHub). See the Nature Portfolio guidelines for submitting code & software for further information.

## Data

Policy information about availability of data

All manuscripts must include a data availability statement. This statement should provide the following information, where applicable:

- Accession codes, unique identifiers, or web links for publicly available datasets
- A description of any restrictions on data availability
- For clinical datasets or third party data, please ensure that the statement adheres to our policy

Raw snRNA-seq and genotype data from the Bryois_192 dataset is available as per their publication at the European Genome-Phenome Archive (EGA) under accession code EGAS00001006345 [7]. Raw snRNA-seq and genotype data from the MATTHEWS dataset is hosted at Synapse under accession code syn54083444. Newly generated raw snRNA-seq and associated genotype data (MRC_60 and Roche_PD) is available under accession code EGAS50000000687. Genotype data is considered personal data and is therefore under protected access by the host repository (EGA), where access is subject to the submission of an application delineating the scope of the project and the data required (full details on the portal). Applications are aimed to be reviewed within two weeks.

Processed single-cell expression counts for each dataset and the full set of eQTL summary statistics for both the full and control-only datasets are available at https://zenodo.org/records/13343729 . The full set of published GWAS summary statistics used for the colocalisation and MR analysis as well as links to the original publications are described in Supplementary Table 3.

## Research involving human participants, their data, or biological material

Policy information about studies with human participants or human data. See also policy information about sex, gender (identity/presentation), and sexual orientation and race, ethnicity and racism.

| | |
|---|---|
| Reporting on sex and gender | Yes, sex is included as clinical covariates for the eQTL mapping. |
| Reporting on race, ethnicity, or other socially relevant groupings | Yes, there is a description of genetic ancestry (white european). |
| Population characteristics | Yes, clinical covariates have been included and contain age, sex, genotypic information, disease diagnosis as assessed by neuropathology. All samples were collected post-mortem. |
| Recruitment | This research was conduceted under the oversight of Imperial College Research ethics. |
| Ethics oversight | Imperial College Research Ethics reference: ICREC_14_2_11 |

Note that full information on the approval of the study protocol must also be provided in the manuscript.

# Field-specific reporting

Please select the one below that is the best fit for your research. If you are not sure, read the appropriate sections before making your selection.

☒ Life sciences ☐ Behavioural & social sciences ☐ Ecological, evolutionary & environmental sciences

For a reference copy of the document with all sections, see nature.com/documents/nr-reporting-summary-flat.pdf

# Life sciences study design

All studies must disclose on these points even when the disclosure is negative.

| | |
|---|---|
| Sample size | We performed snRNA-seq on all brain samples available to us, yielding N=409 individuals (391 post quality control). It is the largest dataset to date with almost equal sizes of controls (N = 183) and disease cases (N = 208), allowing to isolate disease-specific effects of feQTLs at cell-type specific level. |
| Data exclusions | Nuclei with less than 500 UMIs in 300 features, and more than 5% Mitochondrial content were excluded. Related individuals based on genotypic data were excluded, and individuals with less than 10 nuclei for a single cell-type were removed. |
| Replication | Replication was made by comparing eQTL discovery to a large-scale eQTL study performed in bulk brain tissue (N = 6,523). Between 72.9-88.7% of cell-type eQTLs (depending on cell type) replicated at FDR < 5%, of which 90.0–98.3 had the same direction of effect. |
| Randomization | Grouping was done based on diagnosis, determined by neuropathology. eQTL discovery was conducted on the full dataset (N = 391) and on the controls-only dataset (N = 183). No other grouping or selection was made. |
| Blinding | Blinding was not implemented to group allocation. However, in our case, the analysis focused on objective genetic and expression data, where researcher bias is unlikely to influence the outcome. Our study design required knowledge of group allocation to conduct separate analyses for controls and the full cohort, which is standard in genetic studies aiming to capture eQTLs across different biological conditions. |

# Reporting for specific materials, systems and methods

We require information from authors about some types of materials, experimental systems and methods used in many studies. Here, indicate whether each material, system or method listed is relevant to your study. If you are not sure if a list item applies to your research, read the appropriate section before selecting a response.

## Materials & experimental systems

| n/a | Involved in the study |
|-----|----------------------|
| ☒ ☐ | Antibodies |
| ☒ ☐ | Eukaryotic cell lines |
| ☒ ☐ | Palaeontology and archaeology |
| ☒ ☐ | Animals and other organisms |
| ☒ ☐ | Clinical data |
| ☒ ☐ | Dual use research of concern |
| ☒ ☐ | Plants |

## Methods

| n/a | Involved in the study |
|-----|----------------------|
| ☒ ☐ | ChIP-seq |
| ☒ ☐ | Flow cytometry |
| ☒ ☐ | MRI-based neuroimaging |

## Plants

| | |
|---|---|
| Seed stocks | *Report on the source of all seed stocks or other plant material used. If applicable, state the seed stock centre and catalogue number. If plant specimens were collected from the field, describe the collection location, date and sampling procedures.* |
| Novel plant genotypes | *Describe the methods by which all novel plant genotypes were produced. This includes those generated by transgenic approaches, gene editing, chemical/radiation-based mutagenesis and hybridization. For transgenic lines, describe the transformation method, the number of independent lines analyzed and the generation upon which experiments were performed. For gene-edited lines, describe the editor used, the endogenous sequence targeted for editing, the targeting guide RNA sequence (if applicable) and how the editor was applied.* |
| Authentication | *Describe any authentication procedures for each seed stock used or novel genotype generated. Describe any experiments used to assess the effect of a mutation and, where applicable, how potential secondary effects (e.g. second site T-DNA insertions, mosiacism, off-target gene editing) were examined.* |

