## [Peer Review File · Nature Genetics]

Cell state-dependent allelic effects and contextual Mendelian randomisation analysis for human brain phenotypes

Corresponding Author: Professor Michael Johnson

A version of this paper was originally rejected for publication by Nature Genetics, however that decision was reconsidered after appeal by the authors.

Version 0:

Decision Letter:

3rd Mar 2023

Dear Professor Johnson,

First, please allow me to apologise for the delay in returning this decision to you. Thank you for bearing with me.

Your Article entitled "Single-cell Mendelian randomisation identifies cell-type specific genetic effects on human brain disease and behaviour" has now been seen by 3 referees, whose comments are attached. While they find your work of potential interest, they have raised serious concerns which in our view are sufficiently important that they preclude publication of the work in Nature Genetics, at least in its present form.

While the referees find your work of some interest, they raise concerns about the strength of the novel conclusions that can be drawn at this stage. In addition, the advance afforded by your work over Bryois et al. is unclear.

Should further experimental data allow you to fully address these criticisms we would be willing to consider an appeal of our decision (unless, of course, something similar has by then been accepted at Nature Genetics or appeared elsewhere). This includes submission or publication of a portion of this work someplace else.

The required new experiments and data include, but are not limited to those detailed here. We hope you understand that until we have read the revised manuscript in its entirety we cannot promise that it will be sent back for peer review.

If you are interested in attempting to revise this manuscript for submission to Nature Genetics in the future, please contact me to discuss a potential appeal. Otherwise, we hope that you find our referees' comments helpful when preparing your manuscript for resubmission elsewhere.

Sincerely,

Safia Danovi
Editor
Nature Genetics

Referee expertise:

Referee #1: GWAS follow-up to disease mechanisms, MR

Referee #2: complex trait genetics, incl. single cell analyses

Referee #3: functional genomics, neurogenetics incl. single cell analyses

Reviewers' Comments:

Reviewer #1:

Remarks to the Author:

Review: Single-cell Mendelian randomisation identifies cell-type specific genetic effects on human brain disease and behaviour

Overview: Haglund, Zuber, et al report a novel approach to infer causal relationships between cell-type specific gene expression and human brain phenotypes. Their approach builds upon previous methods in transcriptome-wide association studies and Mendelian Randomization (i.e. MR) and is highly topical and likely to be of interest to the statistical genetics/genomics communities. To illustrate the utility of their approach, the authors perform single-cell eQTL calling in post-mortem brain tissue samples from 147 "healthy" donors, followed by instrument selection and downstream MR across 16 brain phenotypes. The manuscript is written moderately well and results presented clearly. With that said I have several comments.

Major Comments:

1. The authors report using canonical marker genes to identify cell types. Expression of genes that are associated with these marker genes in the same regulatory network can be associated with cis-eQTL (eg. trans-eQTL) and the potentially clinical outcome. The authors should comment on, or demonstrate, that marker genes are unlikely to confound downstream association (i.e. cell-type tagging rather than mediated expression).
2. The functional enrichment analysis as reported is underwhelming and the results not presented through a strong enough statistical lens. While I applaud the authors effort in characterizing this extremely important regulatory dataset, it is unclear to what extent 40 (~22%) overlap these annotations by chance, and if there is any significant enrichment. Similarly, a comparative analysis using cross-cell type annotations to further support the importance of cell-type specificity in this characterization would be valuable.
3. The authors report a pseudo replication of a subset of their findings using pQTL data from postmortem brain samples. The original protein data are likely measured in a bulk manner, and also reflect a mixture of regulatory effects across cell types, likely impacting replication to some degree. Can the authors comment on genes replicating using pQTLs as a function of the number of associated cell types using the sc-eQTL data?
4. Relatedly, the authors report on the number of their findings with a positive score from OpenTargets, but is unclear 1) what one should expect (i.e. what is the null or baseline to compare with here), and 2) to what extent does the score provide stronger support for some genes versus others. The authors should briefly clarify what the OpenTargets score represents or how it is calculated in the primary text.

Minor Comments:

1. For eQTL mapping, the authors created a pseudobulk dataset for each individual. Supplementary Figure 1 displays the distribution of cell types for each individual. Can the authors demonstrate that total cell-count is unconfounded from
2. Can the authors provide a summary of genetic ancestry in sample donors under study, preferably first two PCs from PCA on individual genotypes. Or at least provide ancestry information in the supplementary.
3. Add a sentence to define PP.H4 (test hypothesis of both "traits" associated and share a single causal variant)

Reviewer #2:

Remarks to the Author:

Authors constructed a single nucleus RAN-seq eQTL resource of human post-mortem brain ($n > 120$). In the resource, $> 570k$ cells across 8 brain cell types were obtained. Integration with the GWAS genotype data identified $> 320k$ cis-eQTLs. They conducted colocalization and Mendelian randomization (MR) analysis with the GWAS of neuro and psychiatric diseases. The MR analysis identified several colocalized variant. The resource of the study looks fine, but this reviewer does not consider the MR analysis identified causal inference.

1. This reviewer considers that what the authors conducted was not the right way to use MR and infer causality. In general, dozens of instrumental variables are required to robustly infer causality in MR.
2. As described in the manuscript, majority of the loci had only "one variant" per locus for MR, where causality cannot be assessed. This reviewer admit that they identified colocalization with GWAS, but not causality.
3. Authors should assess much wider ranges of the GWAS phenotypes.
4. By-directional MR between eQTL and GWAS is requested to infer directional causality.
5. Comparisons between eQTL and pQTL is fine.
6. Discussions on the identified loci seems redundant.

7. The top colocalization is identified for the HLA genes. But, it would be challenging to estimate expression of the HLA genes in the single cell unless construct the specified analytic pipeline.
8. Authors reported overlap of the colocalized genes with drug targets, but without estimating null hypothesis, it would be difficult to estimate overlap enrichment.

Reviewer #3:

Remarks to the Author:

The manuscript by Haglund et al. performed eQTL analysis in single nuclei RNA-seq data across 9 major brain cell types. They report evidence of causal association between the change in expression of 118 genes, revealing candidate targets for risk mitigation and opportunities for shared preventative therapeutic strategies. Overall, this is an important resource for the community but I have some concerns about analysis. It is also not clear to me how novel this is compared to other recently published snRNA-seq eQTL paper (Bryois et al) on the same subset of the data.

Major comments:

1. It's unclear how many new samples being sequenced here vs. what has been shown before. 60 new snRNAseq data generated? Pls clarify this in the results section. It's rather a meta-analysis of some new data with previously published results in Bryois et al..
2. The authors argue that one of the advantages of their study is that their brain snRNA-seq data were constructed exclusively from the normal controls. However, in neurodegenerative disease, it is not a very unlikely scenario that a risk variant has an effect in a premanifest phase, in which slight neurodegeneration already occurs, to promote the manifestation of symptoms. In this case, focusing only on normal controls is not always advantageous. Could the others include the diseased tissues to increase power in identifying more eQTLs?
3. According to Fig. 3, HLA genes, which locate in the MHC region, were also analyzed in their methods. However, the validity for application of Coloc to the MHC region has not been established. Super strong and wide LD patterns can affect its accuracy. The MHC region should be excluded, or some simulations should be done if they want to include it.
4. The coloc results are confounded by including the MHC and MAPT region. It needs to be carefully checked to make not spurious association due to extended LD in these regions. Fig 3. Highlights large number of genes in the HLA or MAPT region. How many novel eQTL is identified if these regions were excluded? Are there any independent effect or is it single variant affect multiple genes ?
5. Why such a liberal threshold for COLOC (0.5)? How many of the eQTL colocalizes at a stringent threshold (PP4 >0.8)?
6. They showed that eQTL data-based snRNA-seq are a valid proxy of pQTL by evaluating the pQTL significance of the eQTL variants or IV variants (?). However, is colocalization analysis between eQTL and pQTL more straightforward to show that eQTL and pQTL are derived from the same causal variants? This may be too strict since single-cell level eQTL effects will partially overlap with bulk level pQTL.
7. In the intersection with drug targets, they only explored interactions between targeted genes and drugs using some databases; however, whether drugs increase or decrease of the expression of target genes in specific cell types should be considered to enhance the quality of this analysis. Isn't there no databases covering this information? For example, there is data on the effects of perturbation by compounds on genes in specific cell types (ncbi.nlm.nih.gov/geo/query/acc.cgi?acc=GSE92742).
8. How does the eQTLs compare to other brain (Bulk) results? There should be some replication.
9. The summary stats from eQTL should be made available to reviewers. It was difficult to find sumstats from this paper. I know there's a website but it needs to be updated.

Minor points

10. In line 310, the authors say "targeting them may offer a route to simultaneously alleviating the cognitive deficit associated with SCZ whilst reducing risk of the disease itself." However, cognitive deficit is not exactly the same as lower IQ. Furthermore, lower IQ is a risk factor of SCZ that precedes SCZ; thus, directly linking it to cognitive deficit as one of the symptoms of SCZ causes a leap in logic. Similarly, isn't it the case that linking pre-morbid neuroticism to a maladaptive personality trait associated with SCZ?
11. In Supplementary Table 8, the columns of Cell-type and Gene are apparently reversed. In addition, there are extra sheets (Sheet 1-3), which should be removed.
12. Raw data should be deposited as well within the European Genome-phenome Archive.

I suggest that you consider Nature Communications as a suitable venue for your work. To transfer your manuscript there, please use our manuscript transfer portal. You will not have to re-supply manuscript metadata and files, unless you wish to make modifications, but please note that this link can only be used once and remains active until used. For more information, please see our [manuscript transfer FAQ](http://www.nature.com/authors/author_resources/transfer_manuscripts.html?WT.mc_id=EMI_NPG_1511_AUTHORTRANSF&WT.ec_id=AUTHOR) page.

Note that any decision to opt in to In Review at the original journal is not sent to the receiving journal on transfer. You can opt in to [In Review](https://www.nature.com/nature-portfolio/for-authors/in-review) at receiving journals that support this service by choosing to modify your manuscript on transfer. In Review is available for primary research manuscript types only.

Version 1:

Decision Letter:

IMPORTANT: Please note the reference number: NG-A61381R-Z Johnson. This number must be quoted whenever you communicate with us regarding this paper.

16th May 2024

Dear Dr Johnson,

Thank you for asking us to reconsider our decision on your manuscript "Cell state-dependent allelic effects and contextual Mendelian randomisation analysis for human brain phenotypes". I have now discussed your appeal with my colleagues, and we think that you have some valid points. We therefore agree to send the manuscript back to the original reviewers. As such, we invite you to make any final revisions and re-upload all your files.

When preparing a revision, please ensure that it fully complies with our editorial requirements for format and style; details can be found in the Guide to Authors on our website (<http://www.nature.com/ng/>).

Please be sure that your manuscript is accompanied by a separate letter detailing the changes you have made and your response to the points raised. At this stage we will need you to upload:

1) a copy of the manuscript in MS Word .docx format.

2) The Editorial Policy Checklist:

<https://www.nature.com/documents/nr-editorial-policy-checklist.pdf>

3) The Reporting Summary:

(Here you can read about the role of the Reporting Summary in reproducible science:

<https://www.nature.com/news/announcement-towards-greater-reproducibility-for-life-sciences-research-in-nature-1.22062>)

Please use the link below to be taken directly to the site and view and revise your manuscript:

Link Redacted

With kind wishes,

Safia Danovi, PhD
Senior Editor, Nature Genetics
ORCID: 0009-0007-7822-5479

Version 2:

Decision Letter:

29th Jul 2024

Dear Professor Johnson,

First, please accept my sincere apologies for the delay in returning this decision to you. Thank you for your patience.

Your Article, "Cell state-dependent allelic effects and contextual Mendelian randomisation analysis for human brain phenotypes" has now been seen by 3 referees. You will see from their comments below that while they find your work of interest, some important points have been raised by Reviewer #2. We are interested in the possibility of publishing your study in Nature Genetics, but would like to consider your response to their concerns in the form of a revised manuscript before we make a final decision on publication.

We therefore invite you to revise your manuscript taking into account all reviewer comments. Please highlight all changes in the manuscript text file. At this stage we will need you to upload a copy of the manuscript in MS Word .docx or similar editable format.

*2) If you have not done so already please begin to revise your manuscript so that it conforms to our Article format instructions, available

[here](http://www.nature.com/ng/authors/article_types/index.html).

*3) Include a revised version of any required Reporting Summary: <https://www.nature.com/documents/nr-reporting-summary.pdf>

Please be aware of our [guidelines](https://www.nature.com/nature-research/editorial-policies/image-integrity) on digital image standards.

Link Redacted

We hope to receive your revised manuscript within four to eight weeks. If you cannot send it within this time, please let us know.

Nature Genetics is committed to improving transparency in authorship. As part of our efforts in this direction, we are now requesting that all authors identified as 'corresponding author' on published papers create and link their Open Researcher and Contributor Identifier (ORCID) with their account on the Manuscript Tracking System (MTS), prior to acceptance. ORCID helps the scientific community achieve unambiguous attribution of all scholarly contributions. You can create and link your ORCID from the home page of the MTS by clicking on 'Modify my Springer Nature account'. For more information please visit please visit www.springernature.com/orcid.

Sincerely,

Safia Danovi, PhD
Senior Editor, Nature Genetics
ORCID: 0009-0007-7822-5479

Referee expertise:

Reviewers' Comments:

Reviewer #1:

Remarks to the Author:

The authors have addressed my initial comments.

Reviewer #2:

Remarks to the Author:

This reviewer acknowledges the efforts of the authors to update the manuscript, especially in increasing the numbers of the brain samples by 40% and GWAS traits to be assessed. However, this reviewer still has concerns on running MR analyses based on a few number of the variants as instrumental. Enrichment with drug targets were not significant when adjusted with background null distributions. Further functional (or experimental) validation of the findings related to inferred causality would be required. Data availability section may need to be more specified in details.

Reviewer #3:

Remarks to the Author:

I have no further comments. The authors have adequately addressed all of my previous concerns. However, I strongly encourage them to make any new data public, including the accession number, and to release all cell type-specific eQTL summary statistics without any p-value restriction prior to publication. Pls provide accession for European Genome-phenome Archive datasets for both snRNASeq and Genotypes. Additionally, the provided GitHub link is currently non-functional; please ensure it is live upon publication.

Version 3:

Decision Letter:

Our ref: NG-A61381R2

29th Aug 2024

Dear Dr Johnson,

Thank you for submitting your revised manuscript "Cell state-dependent allelic effects and contextual Mendelian randomisation analysis for human brain phenotypes" (NG-A61381R2). It has now been seen again by Reviewer #2 and their comments are below. The reviewers find that the paper has improved in revision, and therefore we'll be happy in principle to publish it in Nature Genetics, pending minor revisions to satisfy our editorial and formatting guidelines.

Sincerely,

Safia Danovi, PhD
Senior Editor, Nature Genetics
ORCID: 0009-0007-7822-5479

Reviewer #2 (Remarks to the Author):

The authors well responded the comments.

NG-2023-A61381: “Single-cell Mendelian randomisation identifies cell-type specific genetic effects on human brain disease and behaviour”

We thank the Reviewers for taking the time to review our original submission (title above) and for allowing us to improve our research. We provide a point-by-point Response to Referees and detail changes to the revised manuscript below.

To reflect the substantial changes and expanded scope of our revised submission we have changed the title of our revised manuscript to: “*Cell state-dependent allelic effects and contextual Mendelian randomisation analysis for human brain phenotypes*”.

Summary of the revised submission:

[1] We have increased the cohort sample size from 128 brains to 409 (391 post-QC) and increased the number of central nervous system (CNS) traits analysed from 16 to 41. Across the 41 CNS behavioural, structural and disease traits we now report 501 putative causal gene-trait associations (PP.H4 >0.8), making this the most comprehensive single-cell causal inference study of brain phenotypes yet reported.

[2] In the revised submission we show (to our knowledge for the first time), that single cell type allelic effects on gene expression in the human brain are dynamic and disease-dependent. We quantify the effect of brain disease on allelic regulation of gene expression in the brain, and establish the importance of taking account of cell state (as well as cell type) in causal inference studies of brain disease.

[3] We show that the effects of brain disease on brain gene expression are not adequately controlled for by adjusting gene expression values for disease status, which has been the standard approach to date.

[4] In keeping with disease-dependent allelic effects on brain gene expression, we establish that colocalisations for brain disease can also be both disease-dependent and independent.

[5] We show that the use of human brain samples with no history of brain disease and normal neuropathology can substantially increase discoverable causal inferences for brain traits (an 18% gain in the number of colocalisations in our current study).

[6] Our control brain cohort has been increased from 128 brains to 183 brains (+ 43%), comprising the largest control brain single-cell RNA-seq dataset yet described. These data provide a unique and important resource for connecting genes to CNS phenotypes unconfounded by disease, enabling enhanced interpretation of disease-associated variants (detailed in the revised submission).

[7] Leveraging this resource, we construct genetic instruments that proxy for pharmacological gene inhibition and activation, and report cell type specific associations and contextual therapeutic directionality for 142 gene-trait pairs, unconfounded by disease.

[8] We apply a principled implementation of single-cell Mendelian randomisation (MR) that combines colocalisation and MR, which has been shown to both limit the risk of confounding by horizontal pleiotropy and increase the likelihood of regulatory approval after clinical trials.

[9] We have expanded our analysis to now include plasma proteomic data from the UK Biobank Pharma Proteomics Project (UKB-PPP), revealing candidate disease modifying drug targets that converge across genetically proxied transcript levels in the brain and corresponding protein levels in human plasma, with the latter representing candidate peripheral biomarkers predictive of CNS outcomes.

[10] We provide a statistical and methodological roadmap for the proper conduct of single cell type Mendelian randomisation analysis seeking to connect molecular exposures to brain disease, and

establish an internationally unique control-brain transcriptional resource for the interpretation of CNS GWAS studies unconfounded by disease status.

[11] To reflect these substantial changes we have revised the title to: “*Cell state-dependent allelic effects and contextual Mendelian randomisation analysis for human brain phenotypes*”.

[12] We include a point-by-point response to the Reviewer's comments on our original submission below. Thank you for considering our revised manuscript.

Response to Reviewers

Reviewer #1

[1] The authors report using canonical marker genes to identify cell types. Expression of genes that are associated with these marker genes in the same regulatory network can be associated with cis-eQTL (eg. trans-eQTL) and the potentially clinical outcome. The authors should comment on, or demonstrate, that marker genes are unlikely to confound downstream association (i.e. cell-type tagging rather than mediated expression).

To address this question, we first assessed whether any of the canonical marker genes used to annotate cell type (excitatory neurons: *SLC17A7*, inhibitory neurons: *GAD2*, endothelial cells: *CLDN5*, microglia: *CIQA*, oligodendrocytes: *MOG*, astrocytes: *AQP4*, oligodendrocyte precursor cells (OPCs): *PDGFRA*, pericytes: *RGS5*) overlapped any genome-wide significant GWAS locus for any of the CNS traits analysed (41 in the revised submission). Across all traits, the only cell type marker genes overlapping GWAS loci were *SLC17A7* (overlapping loci for multiple sclerosis and schizophrenia) and *MOG* (overlapping loci for intelligence, multiple sclerosis, schizophrenia and sleep duration). Across each of these, there were no colocalisations (all PP.H4 probabilities < 0.1, mean PP.H4 = 0.07, median PP.H4 = 0.07). As a further analysis, we considered whether a gene or genes in the same regulatory network as a marker gene could lead to a spurious association between a transcript and an outcome due to confounding by cell type. To assess this, we plotted the residual gene expression of marker genes (following covariate adjustment) against the residual expression of genes inferred to have an association with an outcome by MR. Across all gene-cell-type pairs, only 9 had an absolute correlation above >0.5 (positive or negatively correlated), with mean Pearson correlation (in absolute value) $r = 0.15$, median = 0.13.

[2a] The functional enrichment analysis as reported is underwhelming and the results not presented through a strong enough statistical lens. While I applaud the authors effort in characterizing this extremely important regulatory dataset, it is unclear to what extent 40 (~22%) overlap these annotations by chance, and if there is any significant enrichment.

We thank the Reviewer for this comment and agree that our assessment of the functional variant required a more structured statistical context. The proper conduct of MR requires instrumental variables (IVs) that are robustly associated with an exposure. We selected IVs based on the strength of the statistical association between genetic variation and the level of a transcript in single cell types and then, to ensure the use of independent instruments, we removed all SNPs in linkage disequilibrium (LD) with the lead eQTL SNP. However, because of LD structure, the selected IVs are not necessarily the functional regulatory SNP/s. Rather than considering the epigenetic context of just the selected IV, we therefore first obtained all SNPs in high LD ($r^2 > 0.9$) for the cell types (microglia, neurons and oligodendrocytes) for which we had appropriate epigenetic data (PLAC-seq, ATAC, H3K27Ac, H3K4me3). Using the data from our original submission, excluding the HLA and MAPT regions (see below), the 101 IVs selected for MR were in high LD with 3,164 SNPs, of which 1,764 intersected with one or more epigenetic features (see Table below). In comparison, when using a set of 1,000 random SNPs, we obtained 28,722 SNPs in high LD ($r^2 > 0.9$), of which 6,748 intersected one or more epigenetic

features. Across each modality, IV SNPs or SNPs in high LD were significantly enriched in one or more epigenetic features compared to the random expectation.

	Intersection – IVs	Intersections – random	P-value
Any PLAC-seq contact	1,614 (51.0%)	5,904 (20.6%)	1.32×10^{-276}
Any ATAC peak	105 (3.3%)	383 (1.33%)	1.91×10^{-14}
Any H3K27Ac peak	445 (14.1%)	1,947 (6.78%)	2.13×10^{-41}
Any H3K4me3 peak	164 (2.43%)	346 (1.20%)	3.48×10^{-44}

Please note however, that in the revised manuscript, we have omitted the section on the intersection of IVs with epigenetic features due to the uncertainties associated with assigning the precise functional SNP with this type of analysis. Instead, in the revised manuscript, we have chosen to focus on the replication of exposure-trait associations identified by single cell type eQTL-anchored MR using independent brain and plasma pQTL data (please see revised manuscript for details).

[2b] Similarly, a comparative analysis using cross-cell type annotations to further support the importance of cell-type specificity in this characterization would be valuable.

We again thank the Reviewer for this insightful comment. In the 186 IV-cell-type pairs used in the epigenetic enrichment analysis above (from the original submission), 66 overlapped an epigenetic mark in the correct cell type. However, permuting the cell-type labels 1,000 times to generate an empirical null distribution revealed only a non-significant trend toward cell type specificity as shown in the plot below (p -value = 0.08).

[3] The authors report a pseudo replication of a subset of their findings using pQTL data from postmortem brain samples. The original protein data are likely measured in a bulk manner, and also reflect a mixture of regulatory effects across cell types, likely impacting replication to some degree. Can the authors comment on genes replicating using pQTLs as a function of the number of associated cell types using the sc-eQTL data?

Reviewer 1 is correct that the pQTL data were obtained from post-mortem bulk-tissue brain samples and therefore the protein levels obtained represent an average from all cell types in the sample and their relative levels of expression. For these reasons, variation in the proportions of the different cell types across the sample set may obscure the detection of some pQTLs due to non-genetic variation in a protein's measured level. Bulk brain tissue pQTL data therefore suffer the same limitations as bulk-tissue derived eQTL data.

Within the brain pQTL data used for this analysis (PMID: 33571421), of 2,474 proteins with a significant (FDR < 0.05) pQTL, 1,064 (43.0%) also had a significant single cell type eQTL. Of these eQTLs, only 342 (32.1%) were found in more than one cell type, suggesting that the estimation of protein expression could indeed be obscured by non- or poorly-expressing cell types. Unfortunately, the current state of technology precludes accurate measurement of cell-type specific protein expression at scale in the brain.

In recent months there has been considerable interest in the use of proteomic data from the UK Biobank Pharma Proteomics Project (UKB-PPP, PMID: 37794186) for drug target and peripheral biomarker discovery. The UKB-PPP assayed plasma proteomic profiles for 54,219 genotyped individuals. Although such plasma proteomic data suffer from the same limitations as bulk brain tissue samples, perhaps even more so given that circulating proteins in plasma likely originate from multiple organs as well as multiple cell types, they offer an opportunity to identify plasma-based biomarkers whose relative dynamic change could be predictive of ultimate clinical outcome.

Therefore, in the revised manuscript, we now include a replication of our single cell MR inferences using pQTL data from UKB-PPP. Thus, for each single-cell MR inference, using pQTL data from UKB-PPP we repeated colocalisation analysis and instrumental variable selection in a *cis*-window to each gene (+/- 500Kb from each end of the gene in question), retaining SNPs with FDR < 5% and independent SNPs ($r^2 < 0.01$). As detailed in the revised manuscript, we find 11 out of 112 gene-trait pairs (excluding the HLA and MAPT loci) identified using single-cell eQTL-anchored MR are independently replicated using pQTLs derived from UKB-PPP (IVW *p*-value < 0.05), and therefore representing candidate peripheral biomarkers predictive of CNS outcome.

[4] Relatedly, the authors report on the number of their findings with a positive score from OpenTargets, but is unclear 1) what one should expect (i.e. what is the null or baseline to compare with here), and 2) to what extent does the score provide stronger support for some genes versus others. The authors should briefly clarify what the OpenTargets score represents or how it is calculated in the primary text.

The Open Targets L2G methodology (PMID: 34711957) is based on a machine-learning model that integrates various sources of evidence to connect genetic variation, genes and phenotypes. The starting point for each gene-to-trait inference in L2G is the GWAS summary statistics which are annotated using different approaches including molecular QTL (e.g., eQTL data based on bulk tissue samples) colocalisation, promotor capture HiC, VEP, and variant-gene distance. To train their model the authors utilize a “gold standard” based on databases such as ChEMBL. Gold standard associations are given a value of 1 and random genes 0. The authors apply a supervised learning model to compute the probabilities of a candidate gene being associated with a phenotype. The L2G score is between 0 and 1, with a cut-off >0.5 defined as “more likely”. Limitations to the L2G score therefore include the validity and comprehensiveness of the gold standards, the lack of cell type specificity, the fact that more than one gene may be predicted for a locus, and the absence of a transparent statistical framework that can be tested using appropriate sensitivity analyses.

In our original submission, we used the Open Targets L2G score as an orthogonal estimate of gene-to-disease association, finding that 49% of our MR results were replicated in Open Targets using an L2G cut-off of 0.5. However, as can be seen above, the L2G score has substantial limitations as a “gold standard” for our work, not least the absence of cell type specificity that can mask potential associations

that are only discoverable at a single cell type level. For our re-submission, we have therefore chosen to omit the use of the Open Targets L2G and instead we now focus on replication of our single-cell MR inferences using independent proteomic data in brain and plasma. Although this restricts the number of replications we can attempt due to limitations on the availability of relevant proteomic data, it allows us to apply the same statistical rigour to the replication analyses as to our discovery dataset and also identifies potential peripheral biomarkers predictive of clinical outcome.

Minor comments.

[1] For eQTL mapping, the authors created a pseudobulk dataset for each individual. Supplementary Figure 1 displays the distribution of cell types for each individual. Can the authors demonstrate that total cell-count is unconfounded from [...].

Unfortunately the question we received seems to be missing an end. As such, we have assumed that the Reviewer is asking whether total cell count is biased by any of the adjoining covariates and have carried out a correlation analysis on total cell count versus these. Total cell count did not correlate with any covariate (see correlation matrix below), with the maximum correlation coefficient observed with post-mortem interval (PMI), where there was a negative correlation, ($r = -0.23$). We also highlight that in both our original and revised submissions these covariates were included in the eQTL modelling.

[2] Can the authors provide a summary of genetic ancestry in sample donors under study, preferably first two PCs from PCA on individual genotypes. Or at least provide ancestry information in the supplementary.

In both our original and revised submissions all individuals were of white European ancestry and we provide below a plot of the first two PCs of this study, merged with genotypes from 1000 Genomes.

In the revised manuscript, the methods now clearly reflect the genetic ancestry of the samples. In addition, as suggested by the Reviewer, we have now included the first 5 PCs in the eQTL modelling.

[3] Add a sentence to define PP.H4 (test hypothesis of both “traits” associated and share a single causal variant).

We apologise for not providing a comprehensive definition of PP.H4. The revised manuscript now contains an explanation of both colocalisation and the PP.H4 measure and why it is important.

REVIEWER #2

[1] This reviewer considers that what the authors conducted was not the right way to use MR and infer causality. In general, dozens of instrumental variables are required to robustly infer causality in MR

We acknowledge that the use of pleiotropy-robust MR methods such as MR-Egger, MR-PRESSO, Median-MR (and many more) is not possible with one or few genetic variants as instruments. However, single instrumental variable MR is an entirely appropriate statistical framework for MR. Indeed, the initial instrumental variable design was exclusively based on a single instrumental variable (Identification of causal effects using instrumental variables. Angrist et al., *Journal of the American Statistical Association*, 1996;91:444-455 and Instrumental variables estimation of average treatment effects in econometrics and epidemiology. Angrist JD., *National Bureau of Economic Research Technical Working Paper Series*, 1991; No. 115). Additionally, the initial implementation of MR for genetically anchored causal inference (Mendelian randomization?: can genetic epidemiology contribute to understanding environmental determinants of disease? Davey Smith and Ebrahim., *International Journal of Epidemiology*, 2003;32:1-22) and the theoretical framework for MR (Mendelian randomization as an instrumental variable approach to causal inference. Didelez and Sheehan., *Statistical Methods in Medical Research*, 2007; 16: 309-30) were developed for a single genetic variant to be used as the instrumental variable. Especially, the ratio estimate as defined for example by Didelez and Sheehan (above) is designed for a single genetic variant.

When conducting MR, the precise methodology needs to be tailored to the research question at hand, and in particular in MR the relationship of instrumental variable/s with respect to the exposure/s. Here it is helpful to distinguish between monogenic and polygenic exposures (see for example; Inferring causal relationships between risk factors and outcomes from genome-wide association study data).

Burgess S, Foley CN, Zuber V. *Annual Review of Genomics and Human Genetics*, 2018;19:303-327). Monogenic exposures include the most reliable assessments of causal relationships as there is a clear biological function linking the genetic variant to the exposure and thus there may be greater confidence in the biological relevance of the genetic instrument (see Burgess, Foley, Zuber reference above).

As mentioned above, we acknowledge that the use of pleiotropy-robust MR methods cannot be used to support the ratio estimate with only one or few genetic variants as instruments. We would like to emphasise however that we have implemented a strict filtering of valid IVs by prior colocalization analysis to minimise any potential confounding by linkage disequilibrium (Combining evidence from Mendelian randomization and colocalization: Review and comparison of approaches. Zuber V, et al., *American Journal of Human Genetics*, 2022; 109:767-782). As noted by others, the inclusion of cis-regulated instruments identified by prior colocalisation has the advantage of limiting the likelihood of confounding by horizontal pleiotropy, and target-indication pairs selected on the basis of combined evidence from both colocalisation *and* MR have a higher likelihood of regulatory approval after clinical trials (please see reference Zheng *et al.*, *Nat Genet* 2020 in the revised manuscript).

Generally, the instrumental variable assumptions in MR are untestable. We have consequently adjusted our language in line with the MR guidelines to use a cautious interpretation and now describe the MR effect estimate as the “association of the genetically predicted levels of an exposure with an outcome” (Guidelines for performing Mendelian randomization investigations: update for summer 2023. Burges S, et al., *Wellcome Open Research*. Published online 2023) or, for brevity, (where appropriate) as “putative causal associations”.

[2] As described in the manuscript, majority of the loci had only “one variant” per locus for MR, where causality cannot be assessed. This reviewer admits that they identified colocalization with GWAS, but not causality.

Please see our response above. Concerning colocalization vs causality, as described above, we have adjusted our language in line with the MR guidelines to use a cautious interpretation and now describe the MR effect estimate as the association of the genetically predicted levels of an exposure with the outcome (Guidelines for performing Mendelian randomization investigations: update for summer 2023. Burgess S, et al., *Wellcome Open Research*. 2023).

[3] Authors should assess much wider ranges of the GWAS phenotypes.

We thank Reviewer 2 for this helpful suggestion and in our revised manuscript we now extend the analysis from 16 to 41 distinct CNS traits across a range of disease, behavioural and structural (e.g., hippocampal volume) brain traits. Where more than one GWAS study has been conducted for a trait we chose the largest and most recent GWAS (hence only one GWAS per trait). In the revised manuscript we now report 501 single cell type colocalisations across 30 CNS traits at $PP.H4 > 0.8$. Additionally, in the revised manuscript, we show that the number of colocalisations is highly dependent on the number of genome-wide significant regions ($r = 0.93$), with many GWAS studies being underpowered for integration with eQTL data using colocalisation. We have therefore opted to focus on a carefully curated set of traits relevant to human brain biology which include unambiguous genetic loci for disease/trait susceptibility.

[4] By-directional MR between eQTL and GWAS is requested to infer directional causality.

We thank the Reviewer for this important point. While the use of brain samples from subjects with no history of brain disease and normal neuropathology should, in most scenarios, mitigate the possibility of reverse causation because of the absence of disease effects on gene expression, it is possible that

some samples are in a “pre-manifest” state that could potentially result in reverse-causal associations. However, to formally test the presence of reverse causation through bidirectional MR, the IV selection must be done on the GWAS SNPs which are genome-wide as shown in the Figure below (PMID: 25064373, <https://mr-dictionary.mrcieu.ac.uk/term/reverse-causality/#>). The selected IVs will necessitate calculated effects on the gene, which can only be estimated through a *trans*-eQTL analysis since the majority of GWAS IVs will not be in *cis*- proximity. This requires an adequately powered sample size for *trans*-eQTL analysis which our current brain single-cell study (or any published study) is inadequately powered for, and may not be feasible without substantial global investment in the collection of non-diseased human brain tissue samples. For example, in a study of gene expression in post-mortem brains using bulk brain tissue samples from 8,613 individuals (PMID: 36823318), only 737 *trans*-eQTLs were discovered in comparison to 16,169 *cis*-eQTLs which suggests that even this large a sample size may be underpowered for *trans*-eQTL analysis and consequently bidirectional MR.

In addition, as can be inferred from the Figure above, that for our causal inferences to be due to reverse causality, this would require sufficient cases for a particular phenotype to be enriched in our control cohort to generate an apparent eQTL association driven by disease rather than by a gene regulatory relationship.

[5] Comparisons between eQTL and pQTL is fine.

Thank you. In the revised manuscript we have now also extended this analysis to include an analysis of UK Biobank Pharma Proteomics Project (UKB-PPP) data to prioritise candidate peripheral biomarkers predictive of CNS outcomes and identify candidate peripheral predictive biomarkers for AD, PD and MS among others.

[6] Discussions on the identified loci seems redundant.

We agree that detailed discussion of identified loci is beyond the scope of this manuscript. We have therefore restricted our text to important loci that also illustrate a wider scientific point (for example, the influence of disease state on the causal inference).

[7] The top colocalization is identified for the HLA genes. But, it would be challenging to estimated expression of the HLA genes in the single cell unless constructn the specified analytic pipeline

We agree that accurate evaluation of *HLA* expression is difficult and traditional read mapping pipelines (such as the 10X *Cellranger*) do not account for the complexity of the *HLA* locus. In line with Reviewer 2's comment, in our revised submission we now aggregate results from the *HLA* genes into a single region ("*HLA*-region") and have amended the abstract, text, figures and tables to reflect this change.

[8] Authors reported overlap of the colocalized genes with drug targets, but without estimating null hypothesis, it would be difficult to estimate overlap enrichment.

We thank the Reviewer for this comment and agree that without a representation of the null hypothesis we cannot determine whether we are enriched for genes that have existing drug targets. Therefore to estimate this we intersected the full set of genes available in our dataset (19,847) with both the STITCH and DGIdb datasets. Using data relating to our original submission we had a total of 118 unique genes with an MR association in one or more cell types to one or more traits. With STITCH, we observed that 58 genes overlapped with one or more compounds (49.2%). Using the full set of 19,847 genes, we observed a similar overlap (47.2%) – i.e., no enrichment in our MR genes. We observed a similar pattern with DGIdb; of 118 genes, 35 were found in the DGIdb database (13.7%), which was lower than with the full set of genes (3,815, 19.2%). It is important however to highlight that we do not necessarily expect an enrichment; DGIdb is based on a mixture of literature mining and known gene-drug interactions (PMID: 33237278), and STITCH follows a similar pattern (albeit more comprehensive) which is extended to chemicals (PMID: 26590256). These databases are usually aggregated and rarely context-specific. We appreciate however that these intersections are underwhelming and they are now excluded from the revised manuscript.

REVIEWER #3

[1] It's unclear how many new samples being sequenced here vs. what has been shown before. 60 new snRNAseq data generated? Pls clarify this in the results section. It's rather a meta-analysis of some new data with previously published results in Bryois et al.

We apologise to Reviewer 3 that we were not clear on this point. In our original submission, snRNA-seq data from 60 control individuals was generated in our laboratory, and this was complemented by 87 controls from Bryois *et al* (PMID: 35915177) via collaboration. In the revised manuscript, we have included new additional and previously unpublished data generated on brains donated by subjects with no history of brain disease and no evidence of brain disease on neuropathological examination (an additional 49 brains) as well as brains donated by subjects who had died with a neurological or psychiatric diagnosis (an additional 108 brains), as well as cases from Bryois *et al.*, (an additional 105 brains) expanding the total cohort to 409 subjects (391 post QC and filtering, split into 183 controls and 208 disease cases). Overall, our control brain cohort has been increased from 128 brains to 183 brains (+ 43%), comprising the largest control brain single-cell RNA-seq dataset yet described.

[2] The authors argue that one of the advantages of their study is that their brain snRNA-seq data were constructed exclusively from the normal controls. However, in neurodegenerative disease, it is not a very unlikely scenario that a risk variant has an effect in a premanifest phase, in which slight neurodegeneration already occurs, to promote the manifestation of symptoms. In this case, focusing only on normal controls is not always advantageous. Could the others include the diseased tissues to increase power in identifying more eQTLs.

We thank the Reviewer for this comment which prompted substantial additional re-analysis now detailed in the revised submission. We would like to highlight that we utilize a uniquely well characterised control cohort, based not just on clinical review but also neuropathological inspection. However, we agree that despite normal neuropathology and no history of neurological disease in our

control samples there is a possibility that “pre-manifest” occult brain disease might be present in some samples, particularly in samples from aged subjects. In the revised manuscript we therefore now comprehensively catalogue the effects of both disease and age on allelic regulation of gene expression in single cell types in the human brain. The results of these extensive analyses are summarised below. For details, please see our revised manuscript.

In summary, with respect to disease, using a principled statistical framework, we now show that while increasing sample size by including disease samples does substantially increase the number of eQTLs discovered (and by extension the number of colocalisations), there is also (a) a substantial proportion of eQTLs and colocalisations that are influenced by the presence of brain disease, despite extensive covariate correction including for disease status and, (b) a substantial proportion of eQTLs and colocalisations that are discoverable only in control brain tissue and which are missed in mixed disease-case and control datasets, again despite appropriate statistical adjustments for disease status.

Concerning the potential for occult brain disease to accrue in control samples with aged subjects, we therefore applied the same interaction methodology developed to test the effect of disease on allelic regulation only this time selecting age as the interaction term. Taking the lead *cis*-eQTL SNP for each eGene, we found that 15.3% were significantly better modelled with age as an interaction term (q -value < 0.05). This proportion varied greatly by cell type ranging from 7.9% (292 out of 3698) in excitatory neurons to 45.1% (83 out of 184) in pericytes. Across all age-interaction eQTLs, 10.8–29.4% of these (depending on cell type) overlapped with disease-interaction eQTLs. Since we cannot exclude the possibility that age-interaction eQTLs overlapping disease-interaction eQTLs might reflect occult pathology, we conservatively excluded these loci from the downstream MR analysis.

[3] According to Fig. 3, HLA genes, which locate in the MHC region, were also analyzed in their methods. However, the validity for application of Coloc to the MHC region has not been established. Super strong and wide LD patterns can affect its accuracy. The MHC region should be excluded, or some simulations should be done if they want to include it.

We agree with this comment and we now report HLA results in aggregate in the revised submission.

[4] The coloc results are confounded by including the MHC and MAPT region. It needs to be carefully checked to make not spurious association due to extended LD in these regions. Fig 3. Highlights large number of genes in the HLA or MAPT region. How many novel eQTL is identified if these regions were excluded? Are there any independent effect or is it single variant affect multiple genes ?

In our original submission we reported a total of 10,288 eGenes in the *cis*-eQTL discovery. When excluding the MAPT region and HLA genes we find a total of 10,157 eGenes. For the colocalisations, we reported 402 colocalisations in total, and 272 when excluding the HLA and MAPT loci. We agree that the HLA region is problematic and in the revised manuscript we now report the aggregated HLA region colocalisations. For the MAPT locus, we appreciate the concern which is now explicitly acknowledged but also recognise that previous publications (e.g., PMID: 34762851 and PMID: 35841044) have considered individual genes within the MAPT locus. Since this information may still be of interest to researchers we have continued to report the MAPT locus in aggregate in the main text of the revised submission but also to include the individual genes in the relevant supplementary tables in case of interest to the reader.

[5] Why such a liberal threshold for COLOC (0.5)? How many of the eQTL colocalizes at an stringent threshold (PP4 >0.8).

We acknowledge the concern raised by Reviewer 3 and in the revised submission we now *only* consider colocalisations at $PP.H4 > 0.8$.

[6] They showed that eQTL data-based snRNA-seq are a valid proxy of pQTL by evaluating the pQTL significance of the eQTL variants or IV variants (?). However, is colocalization analysis between eQTL and pQTL more straightforward to show that eQTL and pQTL are derived from the same causal variants? This may be too strict since single-cell level eQTL effects will partially overlap with bulk level pQTL.

We appreciate the Reviewer's concerns regarding the strictness of straight pQTL/eQTL overlap. The methodology behind genetic colocalisation using COLOC is heavily weighted towards a shared causal SNP – if eQTLs do not overlap with pQTLs even when considering SNPs in LD it is unlikely they will colocalise under a single causal variant hypothesis. In addition, COLOC requires the full set of SNPs and their associations with each trait within a region, which unfortunately is not available for the brain proteome data provided by Robins *et al* (PMID: 33571421). In the revised manuscript however, we have extended our analyses to now include the pQTL associations in plasma from the UK Biobank Pharma Proteomics Project (UKB-PPP), for which the full summary statistics were available allowing us to independently perform genetic colocalisations with brain traits. We then support these colocalisations by selecting independent IVs and undertaking MR which for the majority of protein exposures analysed included multiple independent instruments. Please see the revised manuscript for details.

[7] In the intersection with drug targets, they only explored interactions between targeted genes and drugs using some databases; however, whether drugs increase or decrease of the expression of target genes in specific cell types should be considered to enhance the quality of this analysis. Isn't there no databases covering this information? For example, there is data on the effects of perturbation by compounds on genes in specific cell types (ncbi.nlm.nih.gov/geo/query/acc.cgi?acc=GSE92742).

We agree that drug perturbation data could in theory provide information useful to the modulation of candidate drug targets. Unfortunately, no specific datasets measuring the transcriptional responses of individual human brain cell types to compounds currently exist. Additionally, for datasets such as L1000, the drug perturbations were undertaken in cancer cell lines which may have different transcriptional responses to human brain cell types, often included non-physiological dosing, and were based on short-term (hours to days) drug exposures that fail to account the effects of long-term therapy or the intra and inter-cellular homeostatic cellular mechanisms that function in the brain.

[8] How does the eQTLs compare to other brain (Bulk) results? There should be some replication.

In our original submission we included an assessment of the overlap between our single cell type eQTLs and those published from the Metabrain cohort, an independent study conducted on bulk tissue samples from 8,613 brains (PMID: 36823318). We replicated between 71.3-83.6% of associations (FDR < 0.05) depending on cell type, as shown in Supplementary Fig 2 in the original manuscript. In the revised manuscript, between 72.9 and 88.7% of *cis*-eQTLs replicate in Metabrain, and of these 90.0–98.3% (depending on cell type) had the same direction of effect (see Fig. 1 in the revised manuscript). Notably, analysis of *cis*-eQTLs at a single cell type level identified 4,898 more eGenes compared to *cis*-eQTL discovery based on an equivalently sized “bulk” brain tissue analysis based on aggregating counts across all cell types (see Fig. 1D in the revised manuscript), suggesting that the failure to replicate some of our *cis*-eQTLs relates to a failure of bulk-tissue eQTL studies to adequately capture some single-cell eQTLs despite substantially larger sample size.

[9] The summary stats from eQTL should be made available to reviewers. It was difficult to find sumstats from this paper. I know there's a website but it needs to be updated.

We apologize to the Reviewer for not providing this upon submission. We provide here a link to access the full genome-wide eQTL associations in every cell type;

<https://imperialcollegelondon.box.com/s/nwtlx0eitcluqng727sqmkerwo0ep02t>

Please note that the full summary statistics will be made publicly available upon publication of the manuscript.

Minor comments:

[10] In line 310, the authors say “targeting them may offer a route to simultaneously alleviating the cognitive deficit associated with SCZ whilst reducing risk of the disease itself.” However, cognitive deficit is not exactly the same as lower IQ. Furthermore, lower IQ is a risk factor of SCZ that precedes SCZ; thus, directly linking it to cognitive deficit as one of the symptoms of SCZ causes a leap in logic. Similarly, isn't it the case that linking pre-morbid neuroticism to a maladaptive personality trait associated with SCZ.

We apologise that our discussion on this point was not clear. The point of interest we were trying to convey is that genetic effects on behavioural and psychiatric phenotypes may overlap, and that such overlaps may share different directions in terms of the relationship between exposure and trait. In this context, lowered IQ should not be viewed as a risk factor for SCZ, but perhaps rather as an inherent pre-manifest aspect of the disease. The point is well taken, however, and we agree that the terminology around these discussions is unclear. In the revised manuscript, we have chosen simply to report the number of occurrences where a change in the level of an exposure is associated to more than one trait, rather than to speculate on how such effects might translate to therapeutic approaches.

[11] In Supplementary Table 8, the columns of Cell-type and Gene are apparently reversed. In addition, there are extra sheets (Sheet 1-3), which should be removed.

These have been fully corrected in the revised manuscript.

[12] Raw data should be deposited as well within the European Genome-phenome Archive

We will make all raw data available upon publication of the manuscript within the European Genome-phenome Archive.

Response to Reviewers

Re: NG-A61381R1: Cell state-dependent allelic effects and contextual Mendelian randomisation analysis for human brain phenotypes.

Reviewer #1:

The authors have addressed my initial comments.

We would like to thank Reviewer #1 for taking the time to review our revised manuscript and for their comments which have significantly helped to improve the manuscript.

Reviewer #2:

This reviewer acknowledges the efforts of the authors to update the manuscript, especially in increasing the numbers of the brain samples by 40% and GWAS traits to be assessed.

We would like to thank Reviewer #2 for taking the time to review our revised manuscript and for their comments which have significantly helped to improve the manuscript. We are grateful for the Reviewer's positive comments on our efforts to increase sample size. The total number of brains analysed has increased from 128 to 409 (391 post-QC – a 3-fold increase on the previous submission). The number of control brains increased from 128 to 183 (+43%). The number of unique CNS traits analysed increased from 16 to 41.

However, this reviewer still has concerns on running MR analyses based on a few number of the variants as instrumental.

We acknowledge that the Reviewer wishes to see a multi-instrument MR approach. We therefore now additionally implement a multi-instrument MR method (**Burgess et al., *Genetic Epidemiology* 2017, PMID: 2894451**) as a technical validation of our statistical methodology. Using this multi-instrument approach to MR we technically validate 138 out of our original 140 MR associations, using a mean $n_{IV}=33$ independent instrumental variables (IVs) per gene (range 10 – 91).

Before detailing our multi-instrument MR analysis however, we stress that our use of combining colocalisation with carefully selected single/few instruments for MR is the current state-of-the-art for cis and drug target MR, which has been followed by multiple authors in recent high-impact peer-reviewed journals. For example, Yazar et al (***Science*, 2022, PMID: 35389779**) used only a single SNP supported by prior colocalisation as IVs in their single-cell MR analysis of peripheral immune cell types. Considering bulk-level gene expression data as exposures, de Klein et al (***Nature Genetics*, 2024, PMID: 36823318**) conducted a single-SNP MR analysis followed by colocalization. Similarly, a comprehensive MR analysis of GTEx data from Professor George Davey Smith's laboratory (Richardson et al., ***Nature Communications*, 2020, PMID: 31924771**) also used predominantly single IVs in their MR framework (for example, only 285 out of 7865 genes with an eQTL using thyroid tissue had more than a single instrument).

The single-instrument approach has also been followed at a protein level. Considering proteins as exposures, Pietzner et al (***Nature Communications*, 2022, PMID: 35970849**) followed the same procedure with 'colocalisation first' to identify proteins associated with an increased risk of severe Covid-19 infection followed by single-instrument MR. Similarly, Gaziano et al (***Nature Medicine*, 2021, PMID: 33837377**) investigated 1,263 actionable proteins and their impact on the severity of Covid-19 infection where they first performed MR using, on average, 1-3 independent genetic variants as IVs followed by colocalization. Smyth-Byrne et al (***Nature Communications*, 2024, PMID: 38684708**) used a similar approach to map the impact of proteins on cancer risk. The same approach has also been used in the cardiovascular field for example Yao et al (***Nature Communications*, 2018, PMID: 30111768**) and Folkersen et al (***Nature Metabolism*, 2020, PMID: 33067605**). Beyond

Response to Reviewers

cardiovascular disease, Zheng and colleagues (*Nature Genetics*, 2020, PMID: 32895551) employed a phenome-wide MR approach that also employed single-instrument MR for *cis*-pQTLs.

We also note that the analytical approach of using single/few carefully selected genetic instruments combined with colocalization is the recommended method in several high-quality reviews on drug-target and *cis*-MR, for example, Gill et al., *Wellcome Open Res* 2021, PMID: 33644404; Zuber et al., *Am J Hum Genet* 2022, PMID: 35452592; Burgess et al., *Am J Hum Genet* 2023, PMID: 36736292.

We agree with Reviewer 2 however, that there are potentially hundreds of genetic variants in a single colocalised region which could be used to assess the causal relationship between an exposure and an outcome. When conducting *cis*-MR there is an important trade-off: Whilst the use of multiple genetic variants in the analysis can lead to spurious MR estimates and inflated Type 1 error rates, on the other hand if only one or a few carefully selected genetic variants are used then the majority of the data is ignored. In response to Reviewer 2's suggestion therefore, we now implement an additional multi-instrument MR method that takes account of the totality of the data in the *cis*-region of interest (Burgess et al., *Genetic Epidemiology* 2017, PMID: 28944551).

The approach by Burgess *et al.*, (termed PCA-IVW) summarizes the genetic associations of all genetic variants within the region into independent principal components (PCs). Independent PCs explaining 99.9% of the variance of expression are then used as instrumental variables in the MR analysis. Using this approach, we retain an average $n_{IV} = 33$ IVs for each gene (range 10 – 91 per gene exposure).

In revised **Supplementary Table 9** we now include information on the number of genetic variants for each colocalised region and the respective number of PCs selected for use as IVs for each exposure using PCA-IVW. Using this approach we validate 138 of our original 140 MR findings with the same direction of effect as that seen with single/few instrument MR. We now include this analysis in the revised manuscript and have updated the Methods. The revised text in the re-submitted manuscript reads as follows (see tracked changes, Page 10):

“Whilst careful selection of IVs as described above minimises the risk of spurious MR estimates, the application of single instrument MR is sensitive to the particular choice of variants, and ignores the majority of genetic data in the colocalised region. As a technical validation of these results, we therefore implemented the multi-instrument MR method PCA-IVW[16], which takes account of the full set of variants in the colocalised region. Of the 140 significant MR hits, 138 (98.6%) were replicated using PCA-IVW with the same direction of effect (**Supplementary Table 9**).”

Below, we illustrate the implementation of PCA-IVW using two examples.

As a first example, we consider the association between *CRI* expression in oligodendrocytes and Alzheimer's disease (AD) (single IV MR p -value = 1.99×10^{-33} , beta = 0.14). As detailed in our previous submission, this target-trait association was replicated using independent plasma pQTL data from UK Biobank Pharma Proteomics Project (UKB-PPP) (IVW p -value = 2.10×10^{-2} , beta = 0.11, $n_{IVs} = 9$). Applying PCA-IVW to the single-cell gene expression data and the full set of SNPs in the region ($n = 1,520$), the individual SNP effect sizes were decomposed into 36 PCs that explain 99.9% of the variance in *CRI* expression in oligodendrocytes (see **Figure 1A**, below). Using these 36 PCs as instruments yields a highly significant MR association between the change in expression of *CRI* and AD (PCA-IVW p -value < 2×10^{-16} , beta = 0.1) (**Figure 1B**, below), consistent with our single instrument MR result.

Response to Reviewers

Figure 1: PCA-IVW method for CR1/Oligodendrocytes and AD_2022

As a second example, based on single variant MR, we reported a significant association between the change in expression of *GPNMB* in oligodendrocyte precursor cells (OPCs) and genetic risk to Parkinson's disease (PD) (single IV MR p -value = 1.47×10^{-8} , beta = 0.15). This MR association was replicated using independent brain pQTLs (single IV MR p -value = 2.48×10^{-8} , beta = 0.39) and using plasma pQTLs from UKB-PPP (IVW MR p -value = 1.8×10^{-7} , beta = 0.34, $n_{IVS} = 7$). Using the multi-instrument PCA-IVW method, the 1,766 SNPs in the colocalised region were decomposed into 40 principal components explaining 99.9% of the variance (**Figure 2A**, below). Using these PCs as instruments again yielded a highly significant MR association between the change in expression of *GPNMB* in OPCs and PD risk (PCA-IVW p -value < 2×10^{-16} , beta = 0.15) (**Figure 2B**, below).

Figure 2: PCA-IVW method for GPNMB/OPCs and PD_2019

Enrichment with drug targets were not significant when adjusted with background null distributions.

This analysis is not part of the revised submission. In the revised submission, we focus on replicating target-indication pairs using independent proteomic data. Although the number of replications we can attempt is restricted by the availability of data, it avoids the uncertainties attached to the previous enrichment analysis since, for example, there was no prior expectation that *novel* targets identified in our work should be enriched for *known* targets, or that targets based on genetic support for disease-modifying effects (our work) should overlap with targets for symptomatic-only therapies (which still predominate in the CNS pharmacopoeia) (Minikel et al., *Nature* 2024; PMID: 38632401).

Response to Reviewers

In addition, prompted by the helpful comments from Reviewer 2, we now also include the technical replication of our single-cell MR associations using a multi-instrument cis-MR approach (see above).

Further functional (or experimental) validation of the findings related to inferred causality would be required.

Experimental validation of 138 target-indication pairs is unfortunately beyond the scope of the current manuscript (and not typically required for this type of study). It is also important to note that functional or experimental “validation” itself does not guarantee the clinical success of a target – approximately 50% of drugs fail in development due to lack of clinical efficacy despite “proven” target “validity” (PMID: 37359774). Ultimately, the *gold standard* for assessing causality between an exposure and an outcome is the randomised controlled clinical trial (RCT), and it is precisely the translational gap between pre-clinical functional data and human clinical efficacy that motivates our research as a potential future approach to target validation based on human data.

Data availability section may need to be more specified in details.

Regarding data availability (Reviewers 2 and 3), processed single-cell gene expression count data and the full set of eQTL summary statistics for both the full and control-only datasets are now available at <https://zenodo.org/records/13343729>. The full set of GWAS summary statistics used for the colocalisation and MR analysis are detailed in the revised **Supplementary Table 3**. Raw single-cell and genotype data will be made available at the point of publication within the European Genome Phenome Archive.

Reviewer #3:

I have no further comments. The authors have adequately addressed all of my previous concerns.

We would like to thank Reviewer #1 for taking the time to re-review our manuscript and for their helpful comments which have significantly helped to improve the manuscript.

However, I strongly encourage them to make any new data public, including the accession number, and to release all cell type-specific eQTL summary statistics without any p-value restriction prior to publication. Pls provide accession for European Genome-phenome Archive datasets for both snRNASeq and Genotypes. Additionally, the provided GitHub link is currently non-functional; please ensure it is live upon publication.

We are committed to making all data publicly accessible. Please see our response to Reviewer 2 above. The updated GitHub link for code used in these analyses can be found at <https://github.com/johnsonlab-ic/singlecell-MR>.